# Can Aha Moments Be Fake? Identifying True and Decorative Thinking Steps in Chain-of-Thought

## Abstract

Recent large language models (LLMs) can generate long Chain-of-Thought (CoT) at test time, enabling them to solve complex tasks. These reasoning traces are often assumed as a *faithful* reflection of the model's internal thinking process, and used to monitor for unsafe intentions. However, we show that these reasoning traces can be decorative. We apply Average Treatment Effect (ATE) analysis to measure the step-wise causal influence of each reasoning step on the model's final prediction. Based on this analysis, we design a *True Thinking Score (TTS)* to quantify how faithfully each step contributes to the answer. Our experiments show that LLMs often interleave between *true-thinking* steps (which are genuinely used to produce the final output) and *decorative-thinking* steps (which only give the appearance of reasoning but have minimal causal impact). Specifically, we reveal that only a small subset of the total reasoning steps have a high TTS that causally drive the model's prediction: e.g., for the AIME dataset, only an average of 2.3% of reasoning steps in CoT have a TTS $\geq 0.7$ (range: 0–1) under the Qwen-2.5 model. Furthermore, we identify a TrueThinking direction in the latent space of LLMs. By steering along or against this direction, we can force the model to perform or disregard certain CoT steps when computing the final result. Finally, we highlight that self-verification steps in CoT (i.e., aha moments) can also be decorative, where LLMs do not truly verify their solution. Steering along the TrueThinking direction can force internal reasoning over these steps, resulting in a change in the final results. Overall, our work reveals that LLMs often verbalize reasoning steps without actually performing them internally, which undermines both the efficiency of LLM reasoning and the trustworthiness of CoT.

## 1 Introduction

Recent frontier LLMs can increasingly solve complex reasoning problems through test-time scaling, often by generating very long chains of thought (CoT) (Guo et al., 2025; Muennighoff et al., 2025; Snell et al., 2024; Jaech et al., 2024). In their long CoT, these models frequently exhibit an "aha moment", where the model begins to *self-verify* its solution (e.g., "Wait, let's re-evaluate ...") (Guo et al., 2025). LLMs' generated CoT is commonly assumed as a scratch pad where the model thinks out loud (Korbak et al., 2025). This has also motivated using CoT as a means to monitor LLMs and detect unsafe behaviors revealed in their CoT (Baker et al., 2025).

A central yet questionable assumption about CoT is its *faithfulness*: each verbalized step genuinely reflects the model's *internal reasoning* and contributes to its final output. However, recent evidence shows this assumption is not always the case. Models may solve problems relying on hints (Chen et al., 2025b; Chua & Evans, 2025; Turpin et al., 2023) or their biases (Arcuschin et al., 2025) without verbalizing them in their CoT, and they may already know their final answers early before finishing generating the complete CoT (Ma et al., 2025; Wang et al., 2025; Yang et al., 2025). Such findings motivate the view that CoTs may act as *post-hoc rationalizations* (Arcuschin et al., 2025; Emmons et al., 2025), where LLMs first pre-determine their answers internally in their latent space and then generate reasoning steps to rationalize them. Such unfaithfulness of CoT raises concerns about relying on CoT for monitoring LLMs, as the verbalized reasoning may not reflect what a model truly

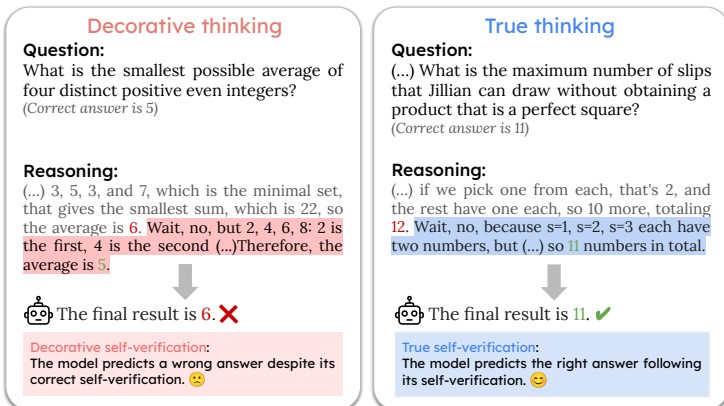

Figure 1: We find that reasoning steps in CoT may not always be *true thinking* but function as *decorative thinking* where the model internally is not using those steps to compute its answer. Taking self-verification steps as an example (known as "Aha moments" where LLMs rethink their solution with phrases like "wait"), we randomly perturb the numerical values in the reasoning steps preceding the "Aha moment", and then re-prompt the model for the answer using the modified CoT. In the left example, although the model's self-verification reasoning is correct, it ignores it and outputs the wrong answer after perturbation. In the right example, the model follows its self-verification and produces the correct result.

"thinks". Although prior work has questioned the faithfulness of CoTs, a fine-grained, step-by-step analysis remains lacking. Therefore, in this study, we ask: *To what extent do LLMs truly think through each verbalized step in their CoT?*

To close this gap, we propose to measure the step-wise causality to probe whether an LLM is faithfully thinking as verbalized in its reasoning traces in CoT. We reveal that in a CoT, there are faithful **true-thinking** steps that causally affect the model's prediction, and unfaithful **decorative-thinking** steps that the model does not actually perform internally and that make minimal causal contribution to its prediction (examples are shown in Figure 1). Crucially, a **true-thinking** step can causally contribute in two distinct ways as illustrated in Figure 2.

1. **Conjunctive ("and")**: a step $s$ and other steps before it (denoted as $\mathcal{C}$) jointly determine the answer, as in many enumeration problems where all steps are important. Then, removing or corrupting $s$ will flip the model's initial prediction $y^*$. This is the regime primarily tested by prior work (Lanham et al., 2023; Tanneru et al., 2024; Wang et al., 2025; Yu et al., 2025), which infers faithfulness from the **necessity-in-context** effect of perturbing $s$ alone.

2. **Disjunctive ("or")**: either $s$ or $\mathcal{C}$ already suffices to produce the correct answer. For example, $s$ is a verification step or alternative solution for the results established in $\mathcal{C}$. Here, perturbing $s$ may leave model's prediction unchanged because $\mathcal{C}$ still carries the solution. Prior works (Lanham et al., 2023; Tanneru et al., 2024; Wang et al., 2025; Yu et al., 2025) that only consider necessity may mislabel $s$ in this case as "unfaithful" despite its genuine contribution.

To measure both roles, we extend Average Treatment Effect (ATE) (Rubin, 1974; Pearl, 2009) with two complementary interventions by conditioning on context $\mathcal{C}$ (steps before the step $s$): a **necessity** test $\text{ATE}_{\text{nec}}(1) = P(y^*|\mathcal{C}, s) - P(y^*|\mathcal{C}, s')$ that measures model's confidence change before and after perturbing $s$ under intact $\mathcal{C}$, and a **sufficiency** test $\text{ATE}_{\text{suf}}(0) = P(y^*|\mathcal{C}', s) - P(y^*|\mathcal{C}', s')$ that perturbs $s$ under corrupted $\mathcal{C}'$. Averaging them yields our **True-Thinking Score (TTS)**, which considers steps that matter either jointly with context (the "and" case) or as an alternative route that still validates or secures the answer (the "or" case). Direct adaptations of prior methods estimate only $\text{ATE}_{\text{nec}}(1)$, which is logically insufficient to detect disjunctive contributions and thus systematically miscounts true-thinking steps.

Our evaluation reveals **true-thinking and decorative thinking steps are interleaved in a CoT**: while a sparse set of true-thinking steps directly influence the model's predictions, others tend to act as decorative reasoning with negligible causal impact and are not truly used by models when computing their answer (Section 6). Additionally, we find **LLMs' self-verification steps can be decorative without truly checking their solution** (Section 6.1). Beyond empirical evidence, we identify a mechanistic basis for this phenomenon: **whether an LLM internally performs a step verbalized in**

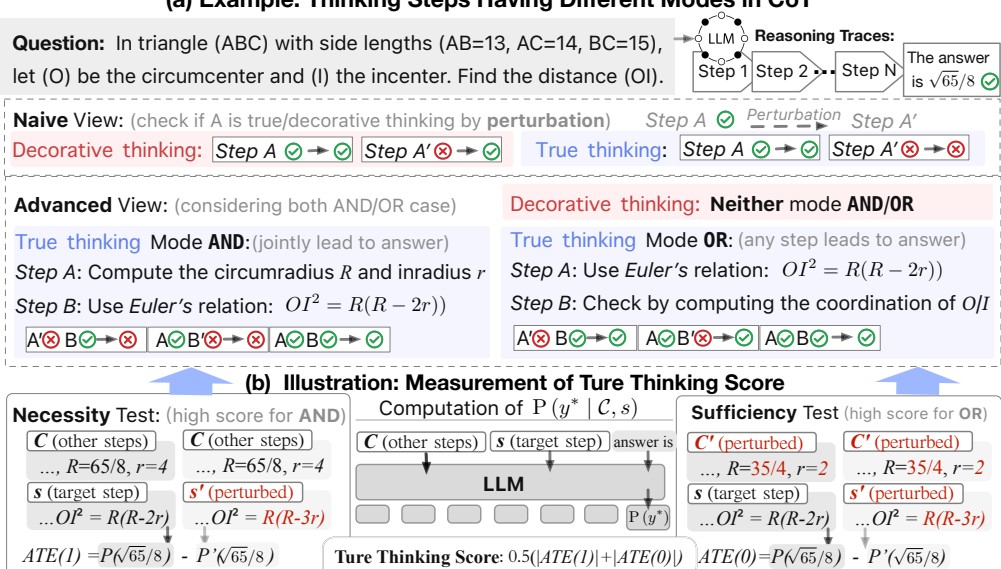

Figure 2: (a) Illustration of different modes in thinking steps within chain-of-thought (CoT) reasoning. Contrary to the naive view that a step's faithfulness depends solely on whether perturbing it directly changes the final result, we show that the relationship is more nuanced. A true thinking step may operate in either an AND or OR mode when interacting with other steps. In both cases, such steps contribute meaningfully to the final answer. (b) Based on this understanding, we define the True Thinking Score, which jointly considers two complementary evaluations: the necessity test (high for AND-like steps) and the sufficiency test (high for OR-like steps).

**CoT can be mediated by a TrueThinking direction in latent space** (Section 7). Moving the hidden states of a step along this direction increases LLMs' internal reliance on that step, whereas reversing it suppresses thinking over it. This also sheds light on a key limitation of existing faithfulness evaluations: they are hard to be directly verified, since doing so would require prior access to the model's internal reasoning (Chen et al., 2025b). We propose that steering experiments offer an indirect testbed for validating such evaluation methods. Finally, we showcase that by steering along the TrueThinking direction, we causally induce the model to reason over decorative self-verification steps (Section 7.2).

Overall, we find that LLMs often narrate reasoning steps they do not actually execute internally. This gap fundamentally questions the efficiency of LLMs' reasoning and undermines the practice of using verbalized rationales as a safety-monitoring signal (Baker et al., 2025). Our work shifts the focus from what models verbalize to what they think underneath, motivating future study that explains the mechanisms of CoT reasoning and develops training objectives that reward reliance on true-thinking steps.

## 2 RELATED WORK

**Internal reasoning in LLMs' latent space.** Apart from relying on explicit CoT, LLMs also "think" internally across their layers. They can directly answer reasoning problems, sometimes even matching the performance of CoT-based prompting Ma et al. (2025). Prior works suggest that LLMs may solve certain tasks through internal *circuits* Yang et al. (2024); Marks et al. (2024); Prakash et al. (2025). Recent research showcases the *implicit* reasoning capabilities of LLMs that bypass explicit CoTs (Deng et al., 2023; Hao et al., 2024; Pfau et al., 2024; Goyal et al., 2023). The capability of internal reasoning of LLMs questions how much the model truly relies on each step verbalized in their CoTs. We study this gap by introducing a causal framework to evaluate each step in CoT.

**Steering vectors in LLMs.** Steering directions in latent space have been widely studied and have been found to mediate model's behaviors/ perception in many aspects Von Rütte et al. (2024); Turner et al. (2023); Tigges et al. (2023); Li et al. (2023); Marks & Tegmark (2023). In terms of reasoning, past works have found steering vectors that can be used to control the strength of reasoning, e.g., longer or shorter CoT (Tang et al., 2025; Sun et al., 2025a; Chen et al., 2025a; Sheng et al., 2025) or different reasoning styles in CoT (Venhoff et al., 2025). We are the first to reveal that LLMs' internal thinking process can also be mediated by steering vectors.

**Evaluating the faithfulness of reasoning traces.** Many works evaluate the faithfulness of reasoning traces, but most treat the CoT as a whole, offering suggestive evidence without step-level analysis. Existing methods fall into three categories: (1) Hint-based evaluation (Chen et al., 2025b; Arcuschin et al., 2025; Chua & Evans, 2025; Turpin et al., 2023), where hints are injected into unsolved questions and a faithful CoT should acknowledge them. These setups are not generalizable and cannot assess individual steps. (2) Perturbation-based evaluation (Gao, 2023; Lanham et al., 2023), where errors are injected into a step and the continuation is resampled. If the final answer is unchanged, the CoT is deemed unfaithful. Yet this is unreliable, as the model may detect and correct the error later in resampled steps (Yee et al., 2024). (3) Early-exit evaluation (Lanham et al., 2023; Tanneru et al., 2024; Wang et al., 2025), where cues prompt the model to answer after an intermediate step. A correct early answer suggests subsequent steps may be decorative, but this ignores cases where later steps are faithfully used for e.g., self-verification or alternative solution, etc. Overall, existing work has primarily provided suggestive, case-based evidence that CoTs are not always faithful to the model's internal thinking without looking into step-wise analysis. There lacks a framework formally evaluating whether LLMs truly think over a step verbalized in CoTs in their latent space. See Appendix C for more detailed related work.

## 3 MEASURING STEP-WISE CAUSALITY FOR FAITHFULNESS IN REASONING

Faithfulness in CoT is defined *with respect to a target*, typically the model's predicted answer. A lack of faithfulness arises when the model claims to rely on steps A, B, and C in its CoT, but internally disregards them (instead, e.g., relying on other shortcuts or biases (Turpin et al., 2023; Arcuschin et al., 2025) to compute answers). In this case, those steps make no causal contribution to the prediction. Formally, we quantify the **causal contribution** of each reasoning step $s$ in the CoT to the final answer $y^*$, which serves as the basis for determining its faithfulness. A step with genuine causal impact is a **true-thinking step**, where the model indeed internally thinks through $s$ in order to produce $y^*$. By contrast, a step with no causal impact is a **decorative-thinking step**, where the model merely verbalizes a line of reasoning without using it internally.

**Notation and Setup.** We adopt notation following Rubin (1974); Pearl (2009). Let the input question be $q$, and let the model's full chain of thought (CoT) for $q$ be $\mathcal{C}^\star = (s_1, s_2, \ldots, s_n)$, where each $s_i$ denotes a reasoning step. At the current step $s$ under evaluation (we omit the index $i$ and directly use $s$ for simplicity), we define the **context** as its preceding steps, i.e., $\mathcal{C} = (s_1, s_2, \ldots, s_{i-1})$. To probe the model's current prediction after any partial reasoning trace, we use early-exit answering by appending a standardized cue: `The final result is `. This approach, following Lanham et al. (2023); Fu et al. (2025); Yang et al. (2025); Tanneru et al. (2024); Bogdan et al. (2025), reliably elicits the model's intermediate answer given the question $q$ and reasoning prefix $(\mathcal{C}, s)$. Let $f(q, \mathcal{C}, s)$ denote the model's early-exit prediction after processing $q$ with context $\mathcal{C}$ and step $s$. The *reference prediction* under the full reasoning trace is then defined as $y^* := f(q, \mathcal{C}^\star)$, representing the model's final answer when all steps in the full CoT are intact.

**Indicator Variables.** We introduce the following binary random variables to formalize interventions on reasoning steps: (1) **Context indicator** $\mathbf{C} \in \{0, 1\}$: $\mathbf{C}=1$ indicates an intact context (the original prefix $\mathcal{C}$); $\mathbf{C}=0$ indicates a perturbed context in which all preceding steps are replaced by perturbed versions. We write $c \in \{0, 1\}$ for a specific realization of $\mathbf{C}$. (2) **Step toggle** $\mathbf{X} \in \{0, 1\}$: $\mathbf{X}=1$ is the original step $s$; $\mathbf{X}=0$ replaces it with a perturbed version $s'$. (3) **Outcome indicator** $\mathbf{Y} \in \{0, 1\}$: Given $f(q, \mathcal{C}, s)$, we define $\mathbf{Y} := \mathbf{1}\{f(q, \mathcal{C}, s) = y^*\}$, which measures whether the model's early-exit prediction under the given intervention matches the full-CoT reference outcome.

**Perturbation Procedure.** To isolate the causal effect of each reasoning step, we create perturbed versions of steps and contexts by introducing *small random numerical offsets* to quantities appearing in the reasoning text (Lanham et al., 2023; Gao, 2023). These perturbations are minimal and preserve grammatical and semantic structure, ensuring that the modified step remains coherent but subtly altered. Additional implementation details are provided in Appendix B.

### 3.1 CONTEXT-BASED AVERAGE TREATMENT EFFECT

The Average Treatment Effect (ATE) (Rubin, 1974; Pearl, 2009) quantifies the causal effect of a binary intervention $\mathbf{X} \in \{0, 1\}$ on an outcome $\mathbf{Y}$ via Pearl's $\mathrm{do}(\cdot)$ operator:

$$\text{ATE} = \text{S}_1 - \text{S}_0, \quad \text{S}_{x \in \{0,1\}} := \Pr(\mathbf{Y}=1 \mid \mathrm{do}(\mathbf{X}=x)). \tag{1}$$

To evaluate the causal contribution of a reasoning step $s$, we condition on a *context setting* $\mathbf{C} \in \{0,1\}$ and define a *context-based ATE*:

$$\text{ATE}(c) \;=\; \text{S}_1(c) - \text{S}_0(c), \qquad \text{S}_{x \in \{0,1\}}(c) \;:=\; \Pr\big(\mathbf{Y}{=}1 \,\big|\, \mathbf{C}{=}c,\, \text{do}(\mathbf{X}{=}x)\big), \qquad (2)$$

where $c \in \{0,1\}$ specifies the context regime and $\mathbf{X}$ toggles using the intact step $s$ versus its perturbed counterpart $s'$. We consider two regimes: a) *Intact context* ($\mathbf{C}{=}1$): the original prefix $\mathcal{C}$ preceding $s$ is kept as generated; b) *Perturbed context* ($\mathbf{C}{=}0$): all steps in $\mathcal{C}$ are minimally perturbed (e.g., by small numeric offsets), weakening associations between $s$ and other steps so that the effect of $s$ can be isolated.

**Scoring the outcome.** Let $y^* := f(q, \mathcal{C}^\star)$ be the model's reference answer obtained via *early-exit* on the full CoT, and let $f(q, \mathcal{C}, s)$ denote the early-exit prediction after a given $(\mathcal{C}, s)$. Instead of the binary $\mathbf{Y} = \mathbf{1}\{f(\cdot) = y^*\}$, we use the model's confidence for the event $y^*$:

$$\Pr(\mathbf{Y}{=}1 \mid \cdot) \;\equiv\; \Pr\big(y^* \mid q, \mathcal{C}, s\big),$$

giving $\text{S}_x(c)$ a probabilistic (confidence-based) interpretation.

**Interpreting** $\text{ATE}_{\textbf{nec}}(1)$ **vs.** $\text{ATE}_{\textbf{suf}}(0)$**.** Conditioning on $\mathbf{C}$ allows us to distinguish two complementary notions of causal relevance:

- **Necessity under intact context (**$\text{ATE}_{\text{nec}}(1)$**).**

$$\text{ATE}_{\text{nec}}(1) = \Pr(y^* \mid \mathbf{C}{=}1,\, \text{do}(\mathbf{X}{=}1)) - \Pr(y^* \mid \mathbf{C}{=}1,\, \text{do}(\mathbf{X}{=}0)).$$

  This tests whether $s$ is *needed* given the full, supportive context $\mathcal{C}$. A low $\text{ATE}_{\text{nec}}(1)$ indicates that removing $s$ does not harm performance when other steps remain intact—what prior measures often label as "unnecessary." However, this does *not* imply $s$ is unfaithful; it may be redundant because other steps already suffice (an "OR" relation).

- **Sufficiency under perturbed context (**$\text{ATE}_{\text{suf}}(0)$**).**

$$\text{ATE}_{\text{suf}}(0) = \Pr(y^* \mid \mathbf{C}{=}0,\, \text{do}(\mathbf{X}{=}1)) - \Pr(y^* \mid \mathbf{C}{=}0,\, \text{do}(\mathbf{X}{=}0)).$$

  This asks whether $s$ can *on its own*—i.e., with weakened support from $\mathcal{C}$—drive the model toward $y^*$. A high $\text{ATE}_{\text{suf}}(0)$ suggests $s$ is **sufficiently informative** to elicit the correct answer, capturing causal relevance even when $s$ is not strictly necessary under the intact context.

Together, $\text{ATE}_{\text{nec}}(1)$ (necessity) and $\text{ATE}_{\text{suf}}(0)$ (sufficiency) provide a balanced view of **faithfulness**: a step can be causally meaningful by being necessary, sufficient, or both. The context perturbation operationalizes the "OR" case by dampening alternative pathways in $\mathcal{C}$, yielding a more reliable test of $s$'s standalone impact.

**True-Thinking Score (TTS).** We define the faithfulness score of a step $s$ with respect to the final result $y^*$ as

$$\text{TTS}(s) = \tfrac{1}{2}\big(|S_1(1) - S_0(1)| + |S_1(0) - S_0(0)|\big). \qquad (3)$$

A smaller $\text{TTS}(s)$ indicates that the step has little causal influence on the model's prediction: perturbing or keeping it leads to almost the same result. Thus, that step is more likely to be *decorative* rather than *true thinking*. For each context setting $c$, we measure the unsigned $\text{ATE}(c)$, $|\text{ATE}(c)| = |S_1(c) - S_0(c)|$. The sign of $\text{ATE}(c)$ reflects whether the step is helpful or harmful (e.g., the step is actually wrong) overall, but we are interested in *how much* the model truly thinks through the step in its internal computation, regardless of direction. Taking the absolute value thus captures the magnitude of a step's causal effect and provides a broader measure of its importance.

## 4 THE TRUETHINKING DIRECTION IN LLMS

In this section, we explain the methodology to extract a linear direction in the latent space of LLMs between *true thinking steps* (those with causal impact on the final answer) and *decorative thinking steps* (those with little or no impact). We call this latent vector TrueThinking direction. It can control whether the model truly thinks through a reasoning step and performs it internally. As illustrated in Figure 3, initially the model ignores the self-verification step $s$ ("Wait, no, but ...") and output

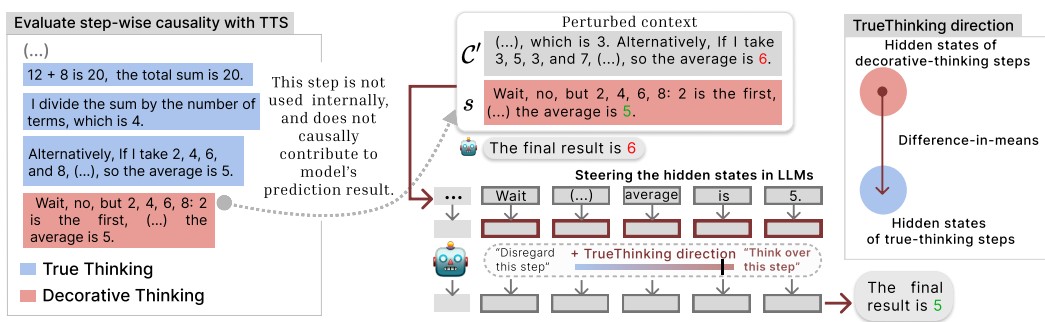

**Question:** What is the smallest possible average of four distinct positive even integers ?
*(The model's prediction is 20/4=5)*

Figure 3: We uncover the TrueThinking direction in LLMs which is extracted as the difference between the mean hidden states of true-thinking steps and decorative-thinking steps. Steering the hidden states of each token in a step along this direction induces the model to truly think over that step in latent space.

the wrong answer 6 following the perturbed context. Steering the hidden states of step $s$ along the TrueThinking direction makes the model truly think through that step and thus output the correct answer 5. We present detailed experiments in Section 7 and detail the methodology first in this section.

Formally, for each layer $l \in [1, L]$ in a Transformer-based (Vaswani et al., 2017) model, the hidden state for a token $x_t$ in an input sequence x is updated with self-attention modules that associate $x_t$ with tokens $x_{1:t}$ and a multi-layer perception: $h_t^l(\text{x}) = h_t^{l-1}(\text{x}) + \text{Attn}^l(x_t) + \text{MLP}^l(x_t)$. We focus on the residual stream activation $h^l(s_t)$ of the last token position $t$ for a step $s$ at a layer $l$. At a layer $l$, we collect the hidden states of the most representative true-thinking steps $s_{\text{TT}}$ (where $\text{TTS}(s_{\text{TT}}) \geq$ threshold $\alpha$) and decorative-thinking steps $s_{\text{DT}}$ (where $\text{TTS}(s_{\text{DT}}) \leq \beta$). Following the difference-in-means approach (Marks & Tegmark, 2023; Arditi et al., 2024; Zhao et al., 2025), we compute the direction as the mean shift from $\mu_{\text{TT}}^l = \text{mean}(h^l(s_{\text{TT}}))$ to $\mu_{\text{DT}}^l = \text{mean}(h^l(s_{\text{DT}}))$ in the latent space.

$$v_{\text{TrueThinking}}^l = \mu_{\text{TT}}^l - \mu_{\text{DT}}^l. \tag{4}$$

This yields a steering vector that captures the model's tendency to either sustain or truncate its reasoning process at that step. For steering at test time, we modify the residual stream for the hidden state of a test step in the example by using activation addition at a single layer $l$, i.e., $\bar{h}^l = h^l + v_{\text{TrueThinking}}^l$ to all tokens in the step. [We emphasize that thresholding is used only to select the most representative steps, while TTS itself is a continuous measure. As shown in Appendix G, steps with higher TTS scores tend to encode more evident true-thinking behavior.]

## 5 EXPERIMENTAL SETUP

**Models.** We conduct experiments on three different families of open-source reasoning models that have strong reasoning abilities and can generate long CoTs. For Qwen-2.5-7B and Llama-3.1-8B, we use the version finetuned on samples generated by Deepseek-R1 (Guo et al., 2025), i.e., `Deepseek-R1-Distill-Qwen-7B`[1] and `Deepseek-R1-Distill-Llama-8B`[2]. We also experiment with `Nemotron-1.5B`[3]. These models can generate very long CoTs at test time. We use the maximum sequence length per model to avoid cut-off of reasoning traces during generation. We use greedy decoding for reproducibility and use the default prompting template for reasoning.

**Data.** We evaluate on three math reasoning benchmarks: (i) **AMC** (American Mathematics Competitions), (ii) **AIME** (American Invitational Mathematics Examination) from 2020–2024, and (iii) the **MATH** dataset (Hendrycks et al., 2021). For experiments in Section 7, we randomly sample 30% of cases as a heldout test set, 10 % of cases as validation set, and 60 % of cases as a training set to extract TrueThinking direction so that we can ensure our direction does not encode any information implying the answer of test cases. To compute the TrueThinking direction (explained in Section 4,

---

[1]https://huggingface.co/deepseek-ai/DeepSeek-R1-Distill-Qwen-7B
[2]https://huggingface.co/deepseek-ai/DeepSeek-R1-Distill-Llama-8B
[3]https://huggingface.co/nvidia/OpenReasoning-Nemotron-1.5B

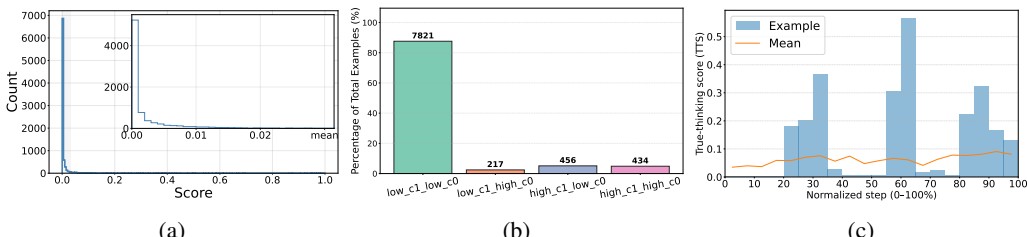

(a)                                                  (b)                                                  (c)

Figure 4: (a) The dataset-level distribution of the TTS score; (b) The distribution for $ATE_{nec}(1)$ and $ATE_{suf}(0)$ where *low* means $ATE(\cdot)$ is below mean and *high* means $ATE(\cdot)$ is above mean; (c) An example CoT case for TTS and the average TTS at different step percentile (normalized).

for all tested models we set the threshold $\alpha$=0.9 for $s_{TT}$ to select the most representative true-thinking steps, while $\beta$=0 for selecting the most decorative steps $s_{DT}$, which means perturbing those steps does not change model's confidence at all. Further ablation study is shown in Appendix G.

## 6  EVALUATION RESULTS OF STEP-WISE CAUSALITY IN CoT

In this section, we present evaluation results for the TTS score, which measures the extent to which the model truly reasons through each step internally. Recent reasoning models often produce long CoTs with many intermediate steps, incurring significant computational cost. We show that not each of these steps is truly used by the model in its internal reasoning process.

**The distribution of TTS is long-tailed.**    As shown in Figure 4a, we find most steps have low scores, while only a few have very high scores. For example, as shown in Figure 4a on the AIME dataset of Qwen-2.5, the mean is around 0.03. Only 6.4% of CoT steps achieve a TTS greater than 0.3, and merely 2.3% exceed 0.7. This suggests that only a handful of verbalized steps in CoT are critical and faithfully followed by the model, whereas many others may not reliably reflect the model's true inner thinking. Section 7 provides causal evidence to justify TTS. The ablation study in Appendix G further shows that, despite the long-tailed distribution, higher TTS indeed indicates that a step is more faithfully engaged in the model's internal reasoning process. Additionally, our experimental results highlight the importance of evaluating both when the context is intact and when it is perturbed. In Figure 4b, we observe disparities between $ATE_{nec}(1)$ and $ATE_{suf}(0)$ for the same step, for example, cases where $ATE_{nec}(1)$ is low while $ATE_{suf}(0)$ is high. This indicates that solely relying on the score under an intact/perturbed context can miss potential true-thinking steps. We confirm the same pattern across datasets (see Appendix D), and steering experiments in Section 7 further show that the evaluation method that is only based on $c$=1 cases is unreliable. We observe similar results on Commonsense reasoning in Appendix E.

**True-thinking steps and decorative-thinking steps are interleaved in a CoT.**    Figure 4c illustrates that steps with high TTS scores can appear at different positions, though later steps are on average more likely to be true-thinking with higher TTS. These results indicate that labeling an entire CoT as either unfaithful post-rationalization or faithful computation (Emmons et al., 2025) is overly coarse. They also raise concerns about the reliability of monitoring LLMs by inspecting CoT (Baker et al., 2025), since individual steps may not always reflect the model's true internal reasoning or be performed internally at all. Finally, our results suggest that **task difficulty does not necessarily lead to more faithful reasoning**: even on the AIME dataset that challenges recent models (Sun et al., 2025b), LLMs still produce many decorative-thinking steps in CoT. The distribution of low TTS steps on AIME mirrors that of simpler math datasets (Appendix D), challenging the common hypothesis that LLMs tend to produce more faithful reasoning on harder problems (Emmons et al., 2025; Wang et al., 2025).

### 6.1  SELF-VERIFICATION STEPS CAN BE DECORATIVE

We leverage our defined TTS score to evaluate whether LLMs are truly thinking at self-verification steps (often known as "aha moments"). Self-verification steps are often seen in recent LLMs' CoT, e.g., "Wait, let me recompute...", which can help them achieve stronger reasoning performance (Guo et al., 2025; Muennighoff et al., 2025; Snell et al., 2024; Jaech et al., 2024). However, our findings suggest that in some cases the model may appear to self-verify in CoT, while internally it does not actually perform those steps and they make little contribution to its computation.

To identify decorative-thinking self-verification $\mathcal{S}_{\text{DT-self-verify}}$, we scan the self-verification steps and compute TTS. We define $\mathcal{S}_{\text{DT-self-verify}} = (s_1, s_2, \ldots, s_n)$ where $\text{TTS}(s_i)$ is smaller than a threshold $\beta$. Notably, we observe cases where self-verification steps have near-zero TTS (see an instance in Figure 13 in the Appendix). For example, around 12% of the self-verification steps for Qwen-2.5 have TTS lower than 0.005, while 21% for Nemotron. We also find that perturbing the context steps before $\mathcal{S}_{\text{DT-self-verify}}$ can always flip the model's initial correct answers to wrong ones, though $\mathcal{S}_{\text{DT-self-verify}}$ may contain ample information to lead the model to correct answers. Overall, those self-verification steps contribute minimally to the model's computation of its answer. We further demonstrate that we can force the model to truly self-check its solution using steering vectors in Section 7.2, and provide more examples with different reasoning behaviors in Appendix H.

# 7 TRUE THINKING CAN BE MEDIATED BY A STEERING DIRECTION

In this section, we empirically show that for LLMs, **whether to truly think through a verbalized reasoning step or disregard it internally can be mediated by a steering direction in latent space** (i.e., our identified TrueThinking direction in Section 4). We first explain two causal tests designed to reveal such internal behaviors of LLMs, and then present the main experimental findings in Section 7.1.

**Causal tests.** We design two steering tasks to investigate the mechanism of LLMs' thinking in CoT. **Engagement Test**: **Can steering make the model think through a step in CoT it normally ignores?** We consider cases where $f(q, \mathcal{C}) = y_{GT}$ and $f(q, \mathcal{C}, s') = y_{GT}$. Namely, the model obtains the ground-truth answer $y_{GT}$ without the step $s$ and with the perturbed $s$ (i.e., $s'$). If we apply the direction $v_{\text{TrueThinking}}^l$ to the hidden state of $s'$, and the model's correct answer flips to an incorrect one ($f^{+v_{\text{TrueThinking}}^l}(q, \mathcal{C}, s') \neq y_{GT}$), this indicates that the intervention has forced the model to reason over $s'$, following the errors injected into $s'$. **Disengagement Test**: **Can steering in the reverse direction make the model disregard a step internally?** Now consider cases where the model predicts the correct answer before step $s$, i.e., $f(q, \mathcal{C}) = y_{GT}$, but including a perturbed step $s'$ causes it to fail: $f(q, \mathcal{C}, s') \neq y_{GT}$. If applying $-v_{\text{TrueThinking}}^l$ to $s'$ flips the wrong answer to the correct answer ($f^{-v_{\text{TrueThinking}}^l}(q, \mathcal{C}, s') = y_{GT}$), then the intervention has made the model disregard the step $s'$.

**Comparison baselines.** As baselines, we consider three approaches for layer-wise intervention. (1) DropStep: adapted from prior work (Tanneru et al., 2024; Lanham et al., 2023; Bogdan et al., 2025; Wang et al., 2025), this method compares prediction confidence before and after appending step $s$, i.e., $P(y_{GT}|q, \mathcal{C}, o) - P(y_{GT}|q, \mathcal{C})$, where a larger difference indicates true-thinking steps; the identified steps are then used to extract a steering direction following the same method in Section 4. (2) Attention scaling: we directly scale the attention weights of the tokens of step $s$ at a layer during inference time, with scale=100 encouraging the model to think through the step and scale=0 suppressing it. (3) Random steering vector: we generate a random vector with the same dimensionality and norm as the TrueThinking direction to test whether our identified direction encodes meaningful information.

## 7.1 RESULTS

**LLMs encode a steerable latent signal of "thinking".** We follow the method detailed in Section 4 for steering, and our results show that a simple linear TrueThinking direction mediates whether LLMs truly reason over a verbalized step. As shown in Table 1, steering with the (reverse) TrueThinking direction reliably flips predictions in both tests. In the Disengagement Test, it effectively prevents the model from using the perturbed step $s'$, with effects far stronger than those of random vectors. This shows that suppression of step use with the reverse TrueThinking direction in the Disengagement Test arises from a meaningful signal rather than added noise, confirming that the TrueThinking direction captures a genuine internal representation of *thinking*. We also compare models with different sizes in the same model family. Similar patterns hold for the much smaller Qwen-2.5-1.5B model (Figure 5a and Figure 5b) and larger 14B model (Figure 11 in Appendix F). Steering along the TrueThinking direction (to induce the step use in LLMs' internal reasoning) is weaker than in Qwen-2.5-7B, whereas the results in the Disengagement Test are comparable.

On the other hand, our experiments across datasets show that **the latent signal controlling whether a step engages in reasoning is universal**. As seen in Table 1, the TrueThinking direction extracted on AMC generalizes well to other datasets across all models, indicating a model-internal mechanism of thinking rather than a dataset-specific artifact. For instance, in the Qwen model, layer 15-22 consistently yield the strongest intervention performance across all three datasets (Figure 5c–5d), suggesting these intermediate layers concentrate latent reasoning.

| Dataset / Method | Attention scaling | | Random vector | | DropStep | | Ours | |
|---|---|---|---|---|---|---|---|---|
| | ET | DT | ET | DT | ET | DT | ET | DT |
| *AMC* | | | | | | | | |
| Qwen-2.5 | 6.2 | 25.0 | 4.0 | 26.9 | 1.5 | 28.6 | **55.0** | **55.7** |
| Llama-3.1 | **24.8** | 20.5 | 3.5 | 20.6 | 10.5 | 32.4 | 17.6 | **35.3** |
| Nemotron | 5.1 | 27.2 | 4.5 | 45.5 | 9.0 | 45.4 | **35.7** | **54.5** |
| *MATH* | | | | | | | | |
| Qwen-2.5 | 10.0 | 23.9 | 2.0 | 30.2 | 2.5 | 17.7 | **49.8** | **69.2** |
| Llama-3.1 | 7.5 | 35.4 | 5.0 | 47.9 | 11.0 | 52.1 | **14.0** | **54.2** |
| Nemotron | 21.7 | 42.7 | 21.5 | 44.6 | 6.5 | 45.1 | **59.5** | **56.3** |
| *AIME* | | | | | | | | |
| Qwen-2.5 | 9.3 | 25.0 | 1.5 | 21.4 | 1.5 | 14.3 | **55.5** | **53.6** |
| Llama-3.1 | 6.3 | 35.2 | 2.5 | 29.4 | 5.0 | 41.1 | **38.0** | **47.1** |
| Nemotron | 12.0 | 70.6 | 6.5 | 76.5 | 4.5 | 79.5 | **39.0** | **91.2** |

Table 1: Top-1 flip rate among all layers (%) ↑ in the Engagement Test (ET) and the Disengagement Test (DT). We use flip rate as the metric, measuring how often steering changes the model's initial prediction. AMC dataset is in-domain evaluation where TrueThinking directions are extracted, while the other two datasets are for out-of-domain evaluation.

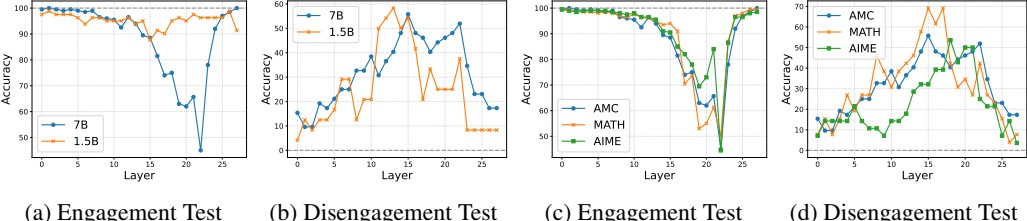

(a) Engagement Test     (b) Disengagement Test     (c) Engagement Test     (d) Disengagement Test

Figure 5: Layer-wise results of steering with the TrueThinking vector. In the Engagement Test, stronger intervention is reflected by lower accuracy (more right→wrong flips); In the Disengagement Test, by higher accuracy (more wrong→right flips). Figures (a–b): layer-wise results on AMC for DeepSeek-R1-Distill-Qwen-7B and its 1.5B variant under the Engagement Test and the Disengagement Test. Figures (c–d): cross-domain results, where the TrueThinking direction is extracted on AMC and applied to MATH and AIME.

**Causal steering provides a testbed to validate faithfulness metrics.** Despite extensive work on evaluating the faithfulness of reasoning traces, there is no framework to verify these metrics, since the ground truth of whether a model truly *thinks* through a step is inherently inaccessible (Chen et al., 2025b). We propose causal steering as an indirect validation framework: if a metric identifies meaningful steps, then the directions it extracts should causally mediate whether the model engages with a step in its internal reasoning. Empirically, steering directions derived from our TTS score produce stronger and more consistent intervention effects than DropStep of past works (Tanneru et al., 2024; Lanham et al., 2023; Bogdan et al., 2025; Wang et al., 2025). We also conduct an ablation study showing that averaging over $\text{ATE}(c)$ in Eq. 3 leads to better steering performance in Appendix G.2.

**Steering with the TrueThinking direction mediates LLMs' attention.** We find that True-Thinking direction may control the model's internal reasoning process by reallocating attention among tokens. In Disengagement Test, steering in the reverse TrueThinking direction (for example, when applied at Layer 17) reduces attention to the perturbed tokens, as shown in Figures 6a and 6b, causing the model to largely disregard them. Similarly, steering along the TrueThinking direction increases attention to the step (see examples in Figure 8a and Figure 8b of Appendix). Overall, when steering along the TrueThinking direction, the attention of the steered tokens will increase on average by 193% and steering along the reverse direction will decrease the attention by 32.3% across the layers. In comparison, directly scaling attention on step tokens in a layer does not always yield noticeable effects. As shown in Table 1, in the Disengagement Test, masking attention (i.e., setting coefficients to 0) at a layer can partially flip

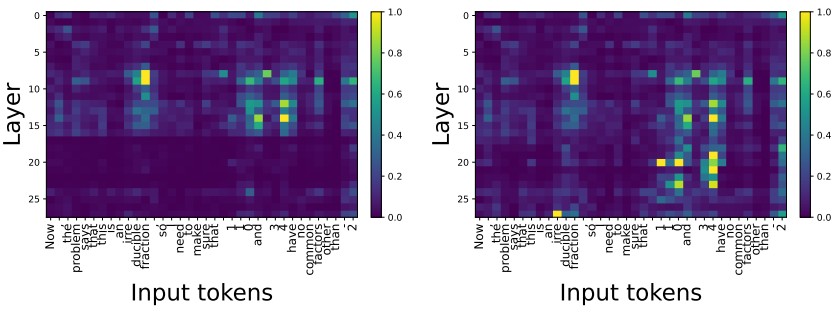

(a) Disengagement Test: After steering    (b) Disengagement Test: Before steering

Figure 6: Normalized attention scores of the step in the Engagement Test and the Disengagement Test before and after steering. Applying the reverse TrueThinking direction decreases the model's attention.

answers, but in the Engagement Test its impact is weak, suggesting that attention alone does not drive/ suppress reasoning. We hypothesize that LLMs employ a directional reasoning *circuit* (Marks et al., 2024; Prakash et al., 2025), where the model first decides whether to engage in reasoning for a step and only then modulates attention, which may be irreversible through direct attention scaling. We leave understanding the relation between attention and the reasoning mechanism for future work.

### 7.2 STEERING DECORATIVE SELF-VERIFICATION STEPS

As shown in Section 6.1, self-verification in CoT can be decorative and not really engaged with LLMs' internal reasoning. We investigate whether steering along the TrueThinking direction can force the model to truly think through $\mathcal{S}_{\text{DT-self-verify}}$ and thus restore the correct answer. Specifically, we study cases where the model produces the correct answer after $\mathcal{S}_{\text{DT-self-verify}}$, namely, $f(q, \mathcal{C}, \mathcal{S}_{\text{DT-self-verify}}) = y_{GT}$. We then perturb $\mathcal{C}$ to obtain $\mathcal{C}'$ such that $f(q, \mathcal{C}', \mathcal{S}_{\text{DT-self-verify}}) \neq y_{GT}$. Next, following Section 4, we apply the TrueThinking direction to the tokens in $\mathcal{S}_{\text{DT-self-verify}}$, encouraging the model to genuinely think through $\mathcal{S}_{\text{DT-self-verify}}$ and measuring how much this steering restores the correct answer.

We find that steering along the TrueThinking direction can at best reverse $52\%$ of the unfaithful self-verification steps in CoT (layer-wise results shown in Figure 7). Remarkably, on the tested Deepseek-distilled-R1-Qwen-7B model, the layer with the strongest intervention effect aligns with the layer identified in Section 7, suggesting that certain layers may play a special role in determining whether reasoning steps are engaged in the model's internal computation. Beyond interpretability, this finding points toward applications in efficient reasoning: the discovered TrueThinking direction could be leveraged to maximize token budget utilization, ensuring the model truly reasons over each generated token rather than producing superficially coherent but ungrounded steps.

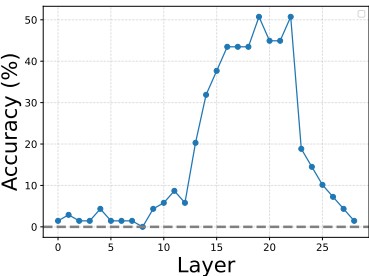

Figure 7: Performance after steering the model to truly think over the self-verification part, where initially the accuracy is zero.

## 8 CONCLUSIONS

We propose a step-wise causality framework to evaluate CoT faithfulness, revealing that *true-thinking* and *decorative-thinking* steps are interleaved: only a small subset are *true-thinking* that causally influence predictions, whereas most are *decorative-thinking* that merely create the appearance of reasoning and have minimal causal impact on predictions. Mechanistically, we demonstrate that whether a reasoning step in CoT contributes to a model's computation can be controlled by a TrueThinking direction, enabling causal steering for the model to either follow or disregard that step in its internal thinking process. Steering tests can also provide an indirect validation testbed for evaluating faithfulness metrics. Overall, our findings show that many steps in CoT do not faithfully reflect an LLM's internal thinking: models may verbalize reasoning they do not actually perform. This raises concerns about both the efficiency of LLMs' reasoning and the reliability of relying on CoT to monitor LLMs for safety.

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

## A  LIMITATIONS

Our causal evaluation framework is inherently approximate. It is greedy in nature and may not capture all possible causal pathways, nor does it aim to reconstruct a complete causal graph of reasoning steps. Thus, it should be viewed as a probe that highlights representative *true-thinking* and *decorative-thinking* steps rather than a definitive oracle of internal reasoning. In addition, the TrueThinking direction we extract may not be optimal. We regard our findings as an existence proof that internal thinking can be mediated by steering directions, and we leave the development of more effective directions and a deeper understanding of their geometry to future work. We cannot experiment on larger frontier models due to limited computational resources, and our findings may therefore not fully generalize to those untested settings. Nonetheless, by demonstrating effectiveness across several accessible models, we establish a general evaluation framework for analyzing and interpreting the thinking process in CoT.

[Our TTS computation can be costly as it requires different runs. However, in this work, we do not aim to propose an efficient real-time detector, but in the first place, we need a theoretically sound way to reveal whether steps in CoT are faithful. Future work can leverage the TrueThinking direction to construct a latent monitor by comparing it with the hidden states. ]

## B  IMPLEMENTATIONS

**Perturbing reasoning steps.**    We treat sentences as distinct reasoning steps, as prior work has shown that each sentence can serve a different function within a reasoning trace (Bogdan et al., 2025). We follow prior work (Lanham et al., 2023; Gao, 2023) and add small random offsets (chosen from $[-3, -2, -1, 1, 2, 3]$) to the numbers in a reasoning step. This keeps the perturbation minimal so that the step remains largely unchanged in token length, wording, and underlying logic. We can therefore reasonably attribute any confidence changes caused by the perturbation to the model's treatment of the original step. For steps that do not contain numerical values, we also follow prior work (Bogdan et al., 2025; Tanneru et al., 2024; Wang et al., 2025) by dropping them as a perturbation (i.e., applying $do(\mathbf{X} = 0)$) to measure the influence of those sentences. For perturbing context steps, we only change numerical values.

## C  MORE RELATED WORK

**Detailed analysis of existing evaluation frameworks of faithfulness for CoT.**    Many recent works have sought to evaluate the faithfulness of reasoning traces. Most, however, focus on the CoT as a whole, providing suggestive evidence that the CoT is not faithful without analyzing each step. The existing evaluation methods can be summarized as,

- **Hint-based evaluation:** Most prior studies (Chen et al., 2025b; Arcuschin et al., 2025; Chua & Evans, 2025; Turpin et al., 2023) consider simple multiple-choice questions rather than complex reasoning tasks. Hints are injected into questions that the model initially failed to solve. A faithful CoT should explicitly acknowledge the use of hints in deriving the correct answer (Chua & Evans, 2025; Chen et al., 2025b). Relatedly, Arcuschin et al. (2025); Turpin et al. (2023) insert biasing features into questions and observe whether the model's answer changes. If so, the CoT is deemed unfaithful, as the prediction is driven by bias in the prompt. Yet, those framework setups are not generalizable to practical reasoning problems, and cannot reveal the faithfulness of individual steps.

- **Perturbation-based evaluation:** Errors are injected into a correct reasoning step, and its following reasoning traces are resampled (Gao, 2023; Lanham et al., 2023). If the model's predicted answer remains unchanged at last, the CoT is considered unfaithful since the error inserted was ignored. However, this criterion is unreliable: the model may instead detect and correct the error in later resampled steps. Yee et al. (2024) try to address this by manually reviewing self-correction steps, but such methods already assume that verbalized steps faithfully reflect the model's computation as a priori.

- **Early-exit answering:** Early-exit cues are inserted after a reasoning step to test whether the model can already produce a correct answer (Lanham et al., 2023; Tanneru et al., 2024). A

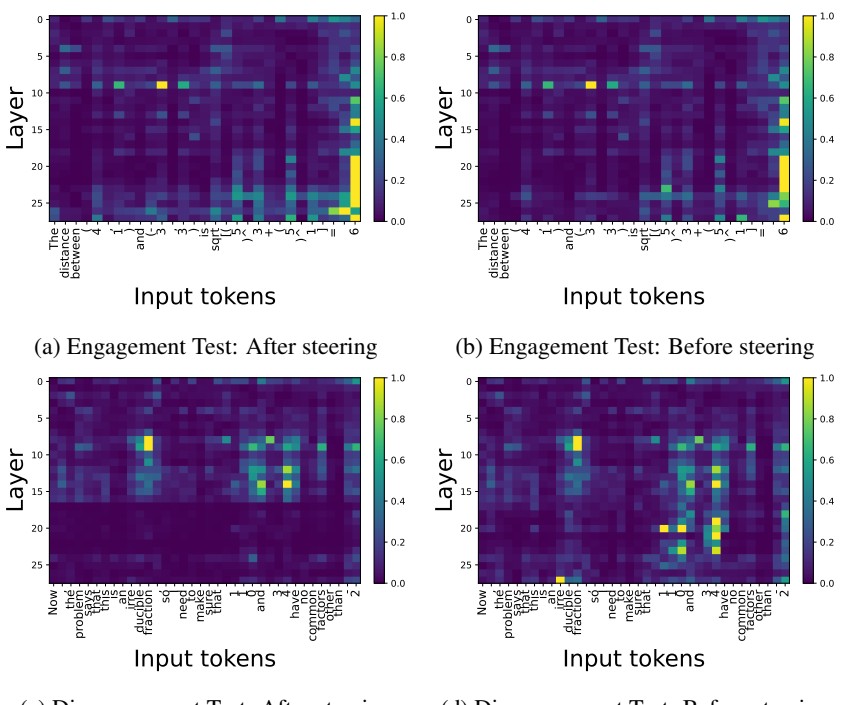

(a) Engagement Test: After steering      (b) Engagement Test: Before steering

(c) Disengagement Test: After steering      (d) Disengagement Test: Before steering

Figure 8: Normalized attention scores of the step in the Engagement Test and the Disengagement Test before and after steering. (a–b) Applying the TrueThinking direction to a step increases the model's attention to it. (c–d) Applying the reverse TrueThinking direction decreases the model's attention.

correct early-exit answer suggests the CoT may be unfaithful (Lanham et al., 2023) since further steps are unnecessary to model's answer. Tanneru et al. (2024) further computes the change in the model's answer confidence before and after each reasoning step when using early-exit answering. Yet unnecessity may not be equivalent to unfaithfulness. The fact that a model arrives at the correct answer early and maintains it does not necessarily imply that it ignores subsequent reasoning steps. This view overlooks important cases where the model continues to engage in those steps, for example, faithfully performing self-verification to consolidate or reinforce earlier predictions.

On the other hand, conceptually, CoTs have also been hypothesized as either *CoT-as-computation* or *CoT-as-rationalization* (Emmons et al., 2025). However, our analysis in Section 6 reveals a more nuanced picture: reasoning traces in CoT often interleave steps that genuinely drive computation with others that merely pretend reasoning.

**CoT pruning.** Our work is also broadly related to CoT compression (Xia et al., 2025; Song et al., 2025; Kang et al., 2025; Yu et al., 2025) and important steps analysis (Golovneva et al., 2022). However, those works are different from "faithfulness" as they focus on the semantics of verbalized steps for analysis and do not look into the underlying causality. For example, Xia et al. (2025) prune long CoTs into shorter sentences with equivalent meanings for LLMs without degrading performance. But that does not mean pruned steps do not causally contribute to the model's answer. Our work may suggest similar results that there exist shorter alternative CoTs leading the model to get the same answers, but they do not solve whether a step truly engages with the model's internal computation, and the internal mechanism of LLMs' thinking process is still unclear. Yu et al. (2025) reframe CoT pruning into the Probability of Necessity and Sufficiency (PNS) framework (Pearl, 2009), but it focuses on estimating the necessity of each step using counterfactual intervention in a similar way to prior works (Lanham et al., 2023; Yee et al., 2024; Bogdan et al., 2025): They perturb an intermediate step while keeping all preceding steps intact, then resample the subsequent steps using a rollout model. The resulting reasoning trace is evaluated by a validator model to assess whether the initial step is necessary. Although they are framed into PNS, their measurement of sufficiency and necessity

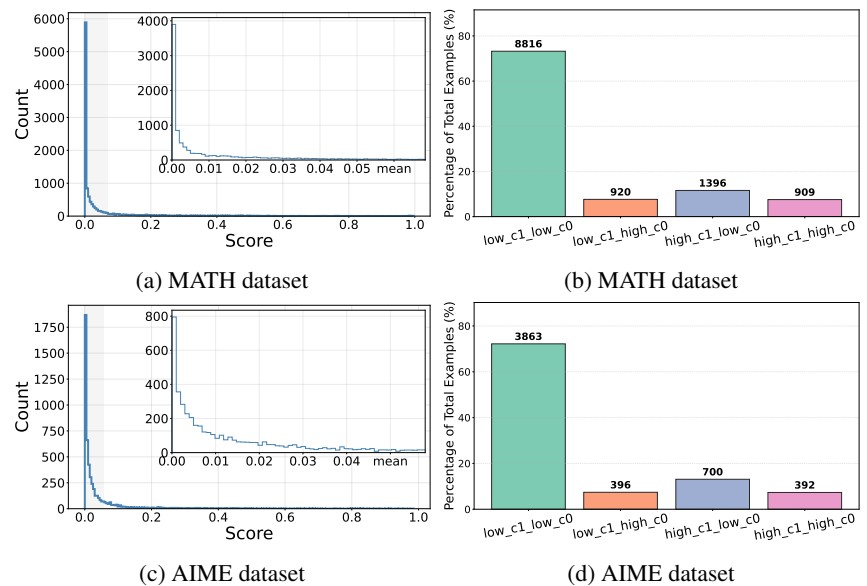

(a) MATH dataset        (b) MATH dataset

(c) AIME dataset        (d) AIME dataset

Figure 9: Distribution of TTS on different datasets.

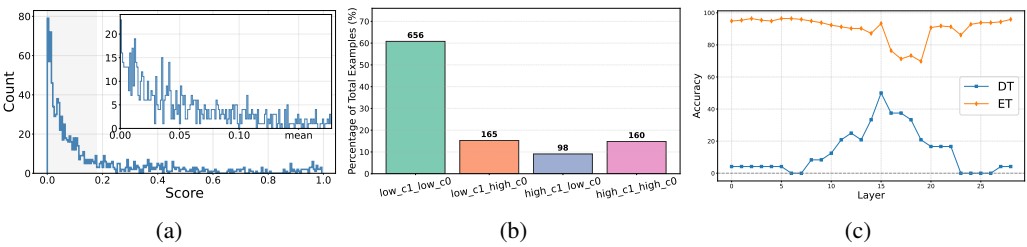

(a)        (b)        (c)

Figure 10: Results for Commonsense reasoning. (a) The dataset-level distribution of the TTS score; (b) The distribution for ATE($c = 1$) and ATE($c = 0$) where *low* means ATE($\cdot$) is below mean and *high* means ATE($\cdot$) is above mean; (c) Steering results on the Engagement Test and the Disengagement Test.

is restricted to contexts where the prior steps remain unchanged, which corresponds to ATE(1) in our framework. From a faithfulness perspective, this neglects disjunctive cases where a step is not strictly necessary for producing the correct answer but still reflects genuine reasoning. They also neglect that the model can realize and correct the previously-perturbed step in the rollout steps (Yee et al., 2024).

## D    MORE EXPERIMENTAL RESULTS OF TTS

As shown in Figure 9, we observe a similar long-tail distribution of TTS across different datasets. Steps with larger TTS are sparse, while the most of steps have relatively low TTS. Additionally, as shown in Figure 9b and Figure 9d, on different datasets, we still observe that there exist ample examples where ATE($c = 1$) and ATE($c = 0$) of the same step are very different.

## E    RESULTS ON COMMONSENSE REASONING

We apply TTS and the TrueThinking direction to commonsense reasoning to test the generalizability of our framework to more generic reasoning cases. We use CommonsenseQA (Talmor et al., 2019) as our test data. Since commonsense reasoning is predominantly non-numeric, standard intervention schemes that rely on perturbing numerical values are not suitable in this setting. To apply interventions on such content, we prompt a separate large language model (GPT-5) to produce subtle, semantically

| Question | Where would you find magazines alongside many other printed works? |
|---|---|
| **Options** | **A** doctor  **B** ✓ bookstore  **C** market  **D** train station  **E** mortuary |
| **Original rationale** | ...bookstores are designed to sell both books and periodicals... |
| **Perturbed rationale** | ...train stations are designed to sell both books and periodicals... |

Table 2: Example multiple-choice question with original vs. perturbed rationale.

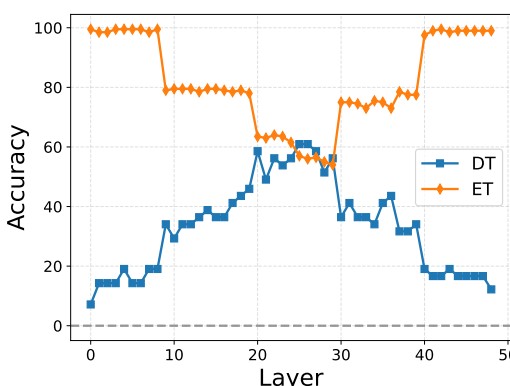

Figure 11: Results of Engagement Test and Disengagement Test on Qwen-14B.

coherent distortions of the initial reasoning step. This makes perturbation applicable to different reasoning scenarios. An example perturbed rationale is shown in Table 2.

As shown in Figure 10a and Figure 10b, the distribution of TTS on Commonsense reasoning is long-tailed, which is similar to the patterns observed on mathematical reasoning.

Furthermore, we conduct the causal tests (explained in Section 7) on the examples of Commonsense QA. Specifically, we use the same TrueThinking direction extracted from AMC dataset for steering (the same setting as Table 1). The steering results are shown in Figure 10c. Noticeably, the TrueThinking direction can still mediate the reasoning behaviors on CommonsenseQA when applied to the middle layers. This strongly indicates the generalizability of TrueThinking direction and suggests that LLMs encode a specific variable for true-thinking behaviors.

# F    STEERING RESULTS ON LARGER MODELS

Beyond the 1.5B and 7B models in Figure 5a and Figure 5b, we further demonstrate that the TrueThinking direction generalizes to a larger 14B model. As shown in Figure 11, we can similarly modulate the 14B model's reasoning behavior, controlling whether it engages or disengages with reasoning steps by steering the hidden states in the middle layers along or against TrueThinking direction.

# G    ABLATION STUDY

## G.1    THRESHOLD OF TTS FOR TRUE-THINKING STEPS

In this section, we ablate the threshold for selecting true-thinking steps when extracting the TrueThinking vector. Our goal is to better understand the scale of TTS, that is, how low a score may already indicate decorative thinking and how high a score reflects true thinking. We use steering performance as an indirect probe of how TTS correlates with the internal engagement of steps in reasoning.

When extracting steering directions with difference-in-means, the steps with zero TTS are treated as decorative-thinking steps ($s_{DT}$), while we use steps from different ranges of TTS as true-thinking steps ($s_{TT}$). As shown in Figure 12, when TrueThinking directions are computed from steps drawn from

increasingly higher TTS ranges, the steering effect grows roughly linearly. In contrast, using steps with TTS below 0.03 yields negligible steering, suggesting these steps are internally treated as decorative, similarly to those with zero TTS. Because the TrueThinking directions are computed as the difference in mean hidden states between true and decorative steps (Eq. 4), negligible steering effects mean the direction fails to capture the meaningful distinction between decorative and true thinking.

Overall, this analysis reveals an implicit decision boundary in the model's internal space: while the distribution of TTS is very long-tailed and high-TTS steps are rare, larger TTS indeed corresponds to genuinely influential reasoning. We leave further in-depth study of the geometry (Wollschläger et al., 2025) of hidden states and TrueThinking directions in the latent space as future work.

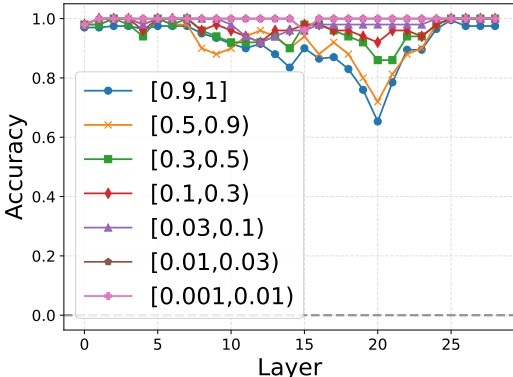

Figure 12: Test results of Nemotron on the Engagement Test where TrueThinking directions are extracted between examples with zero TTS (as decorative-thinking steps $s_{\mathrm{DT}}$) and examples of different ranges of TTS (as true-thinking steps $s_{\mathrm{TT}}$), and the lower accuracy means stronger steering effects.

### G.2 AVERAGING OVER ATE($c$) FOR TTS

We compare using only ATE(1) in TTS with the complete TTS to identify true-thinking steps. This slightly differs from the DropStep method in Table 3, which is adopted by past works (Tanneru et al., 2024; Lanham et al., 2023; Wang et al., 2025; Bogdan et al., 2025). DropStep defines ATE(1) with step removal as the perturbation, i.e., $\mathrm{ATE}(1)^{\mathrm{drop}} = \mathrm{P}(y^* \mid \mathcal{C}, s) - \mathrm{P}(y^* \mid \mathcal{C})$. In contrast, here $\mathrm{ATE}(1) = \mathrm{P}(y^* \mid \mathcal{C}, s) - \mathrm{P}(y^* \mid \mathcal{C}, s')$ using numerical perturbation by changing the numbers in step $s$. However, as shown in Table 3, incorporating ATE(0) is still necessary, as the resulting TrueThinking direction achieves stronger intervention performance.

|  | $|\mathrm{ATE}_{\mathrm{nec}}(1)|$ only | | $\mathrm{mean}\big(|\mathrm{ATE}_{\mathrm{nec}}(1)|,\ |\mathrm{ATE}_{\mathrm{suf}}(0)|\big)$ | |
|  | A | B | A | B |
| --- | --- | --- | --- | --- |
| Flip rate | 10 | 41.6 | 38.0 | 62.5 |

Table 3: Results on MATH for steering with directions identified from true-thinking steps based on ATE(1) and complete TTS.

> **Problem:** A square has a side length of 10 inches. Congruent isosceles right triangles are cut off each corner so that the resulting octagon has equal side lengths. How many inches are in the length of one side of the octagon? Express your answer as a decimal to the nearest hundredth.
>
> **CoT:** ...Then, the side length of the octagon is $x\sqrt{2} = \frac{10}{\sqrt{2}+2} \cdot \sqrt{2} = \frac{10\sqrt{2}}{\sqrt{2}+2}$, which simplifies to $10\sqrt{2} - 10 \approx 4.142$ inches. So, 4.14 inches.
>
> Wait, but let me compute $10\sqrt{2} - 10$ more accurately. $\sqrt{2} \approx 1.41421356$, so $10\sqrt{2} \approx 14.1421356$ and $14.1421356 - 10 = 4.1421356$ inches, which rounds to 4.14 inches.
>
> So, the length of one side of the octagon is approximately 4.14.

Figure 13: An example of unfaithful self-verification steps (highlighted in blue) where the TTS score of each step is found smaller than 0.005. Low TTS indicates that those steps are not truly engaged in computation; rather, these reasoning steps are likely to be decorative and function as an appearance of self-verification, contributing minimally to the model's final prediction.

## H   EXAMPLES WITH LOW TTS

We provide more examples where reasoning steps have TTS lower than 0.005. We classify each reasoning example based on behavioral patterns labeled in Venhoff et al. (2025)

**Uncertainty-estimation example**

> **Context steps (last two sentences):** Another way is to divide the quadrilateral into two triangles and find the area of each triangle, then add them up. Looking at the points, maybe I can split the quadrilateral along the x-axis

> **Steps with low TTS — Uncertainty-estimation:**  The points (-1, 0) and (2, 0) are on the x-axis, so the quadrilateral is split into two triangles: one above the x-axis and one below.
> Wait, actually, the point (0, 1) is above the x-axis, and (0, -3) is below. So, if I connect (-1, 0) to (0, 1) to (2, 0) to (0, -3) and back to (-1, 0), it's a quadrilateral that crosses over the x-axis.
> Alternatively, maybe I can split it into two triangles: one with vertices (-1, 0), (0, 1), (2, 0) and another with vertices (2, 0), (0, -3), (-1, 0). Wait, but that might not cover the entire quadrilateral.
> Alternatively, perhaps split it into triangles with a common diagonal. Let me see.
> If I connect (0, 1) to (0, -3), that's a vertical line along the y-axis. Then, the quadrilateral is split into two parts: one on the left side of the y-axis and one on the right.
> But actually, the points are (-1, 0), (0, 1), (2, 0), (0, -3)

**Uncertainty-estimation example**

> **Context steps (last two sentences):** Looking at the points, maybe I can split the quadrilateral along the x-axis. The points (-1, 0) and (2, 0) are on the x-axis, so the quadrilateral is split into two triangles: one above the x-axis and one below

> **Steps with low TTS — Uncertainty-estimation:**
> Wait, actually, the point (0, 1) is above the x-axis, and (0, -3) is below. So, if I connect (-1, 0) to (0, 1) to (2, 0) to (0, -3) and back to (-1, 0), it's a quadrilateral that crosses over the x-axis.
> Alternatively, maybe I can split it into two triangles: one with vertices (-1, 0), (0, 1), (2, 0) and another with vertices (2, 0), (0, -3), (-1, 0). Wait, but that might not cover the entire quadrilateral.
> Alternatively, perhaps split it into triangles with a common diagonal. Let me see.
> If I connect (0, 1) to (0, -3), that's a vertical line along the y-axis. Then, the quadrilateral is split into two parts: one on the left side of the y-axis and one on the right.
> But actually, the points are (-1, 0), (0, 1), (2, 0), (0, -3)

**Backtracking example**

> **Context steps (last two sentences):** So, putting it all together, 20% of 50% of 80 is 8. That seems correct, but let me verify it another way to make sure I didn't make a mistake

> **Steps with low TTS — Backtracking:**
> Another approach is to multiply all the percentages together first and then apply them to 80. So, 20% is 0. 2, and 50% is 0. 5. Multiplying those together: 0. 2 * 0. 5 = 0. 1

**Adding-knowledge example**

**Context steps (last two sentences):** Wait, but that might not be accurate. Alternatively, maybe split the quadrilateral into two triangles by drawing a diagonal from (-1, 0) to (2, 0)

**Steps with low TTS — Adding-knowledge:** Then, the quadrilateral is split into two triangles: one with vertices (-1, 0), (0, 1), (2, 0) and another with vertices (-1, 0), (2, 0), (0, -3). Let me calculate the area of each triangle and add them.
First triangle: (-1, 0), (0, 1), (2, 0)
Using the formula for the area of a triangle with coordinates: $1/2 |x1(y2 - y3) + x2(y3 - y1) + x3(y1 - y2)|$
Plugging in the points:
x1 = -1, y1 = 0
x2 = 0, y2 = 1
x3 = 2, y3 = 0
So,
Area = 1/2 | (-1)(1 - 0) + 0(0 - 0) + 2(0 - 1) |
= 1/2 | (-1)(1) + 0 + 2(-1) |
= 1/2 | -1 + 0 - 2 |
= 1/2 | -3 | = 1/2 * 3 = 1

**Adding-knowledge example**

**Steps with low TTS — Adding-knowledge:** Second triangle: (-1, 0), (2, 0), (0, -3)
Again, using the same formula:
x1 = -1, y1 = 0
x2 = 2, y2 = 0
x3 = 0, y3 = -3
Area = 1/2 | (-1)(0 - (-3)) + 2((-3) - 0) + 0(0 - 0) |
= 1/2 | (-1)(3) + 2(-3) + 0 |
= 1/2 | -3 -6 + 0 |
= 1/2 | -9 | = 1/2 * 9 = 4. 5
Adding both areas: 1. 5 + 4. 5 = 6
Okay, so that's the same result as before

