# OpenReview forum: "Can Aha Moments Be Fake? Identifying True and Decorative Thinking Steps in Chain-of-Thought"
_ICLR.cc/2026/Conference — Submitted to ICLR 2026_

### Official Review · Reviewer_GQsZ · 2025-10-28

**Soundness:** 3
**Presentation:** 3
**Contribution:** 3
**Rating:** 6
**Confidence:** 4

**Summary:**

This paper investigates the faithfulness of Chain-of-Thought (CoT) reasoning in Large Language Models (LLMs). The authors introduce a novel distinction between "true-thinking steps," which causally influence the model's final prediction, and "decorative-thinking steps," which resemble reasoning but have minimal causal impact. Key findings indicate that only a small fraction of steps in a CoT are "true-thinking". The authors also find that self-verification steps ("Aha moments") can often be decorative. Mechanistically, the paper identifies a "True Thinking direction" in the model's latent space, which can be used to causally steer the model to either utilize or disregard a specific reasoning step.

**Strengths:**

1. The paper addresses a critical and timely question regarding the faithfulness of LLM reasoning. The distinction between "true" and "decorative" thinking provides a valuable new lens through which to analyze and question the internal processes masked by seemingly coherent CoT rationales.
2. The proposed True Thinking Score (TTS) is methodologically sound. Grounding the metric in causal inference (specifically, a context-based ATE) is a significant step beyond simpler perturbation or correlation analyses.
3. A primary strength of the paper is the discovery of the "True Thinking direction." This finding moves the work from a purely observational analysis to one that includes causal intervention. Demonstrating the ability to steer the model's internal reliance on a step provides compelling evidence for the paper's claims and opens up new avenues for model interpretability and control.
4. The paper is well-written, and the core concepts are illustrated effectively. Figures 1 and 2, in particular, do an excellent job of clarifying the concepts of decorative thinking and the TTS evaluation framework. The experimental results are presented clearly and sufficiently support the main claims.

**Weaknesses:**

1. Limitation and Generalizability of the Method: The methodology appears to be designed for and evaluated on tasks with clearly demarcated, sequential, and often computational steps (i.e., mathematical reasoning). It is unclear how this framework would apply to more holistic, ambiguous, or non-linear reasoning tasks (e.g., creative writing, legal analysis, or ethical deliberation) where "steps" are not as discrete and the notion of a single correct "computation" is ill-defined. The authors should discuss these limitations and the potential challenges of generalizing their framework.

2. Interpretation of "Decorative" Steps: The finding that reasoning contribution is unevenly distributed is intuitive. However, the paper's binary classification of steps as "true" vs. "decorative" may be an oversimplification. Steps with low TTS scores are not necessarily "fake" or "decorative" but could be foundational, contextual, or "less important" procedures that are nonetheless part of the necessary logical flow, even if they don't individually have a high causal impact on the final answer. The paper would be strengthened by a more nuanced discussion of this spectrum, rather than a strong binary claim.

3. Computational Cost: The proposed TTS evaluation framework seems computationally prohibitive. It requires multiple perturbed forward passes for every step being evaluated within a CoT. This high cost could severely limit its practical utility for large-scale evaluation, real-time monitoring, or integration into a training pipeline. A discussion of this cost and potential avenues for approximation or optimization would be beneficial.

4. Practical Implications: While the analysis is fascinating (especially Section 7), the paper is light on the practical implications of its findings. The discovery of the "True Thinking direction" is a powerful result, yet the discussion of its applications is underdeveloped. The authors should elaborate on how this insight could be leveraged, for example, to improve model efficiency, enhance faithfulness during fine-tuning, or build more reliable monitoring tools.

**Questions:**

1. Process supervision and Process-based Reward Models (PRMs) are critical for training more reliable reasoning, but they often struggle with providing accurate, fine-grained rewards. Could the True Thinking Score (TTS) be adapted to serve as a more robust reward signal? For instance, could a model be fine-tuned to maximize the average TTS of its reasoning steps, thereby penalizing the generation of "decorative" steps and encouraging more efficient, faithful reasoning?

2. As mentioned in the weaknesses, the experiments are confined to mathematical tasks. How do the authors envision this method being extended to other domains? What new challenges arise when applying TTS to commonsense reasoning, summarization, or dialogue, where steps are less formal and "truth" is more subjective?

3. Beyond its analytical value, what are the concrete practical applications of the identified "True Thinking direction"? Could this vector be used, for example, as a tool to dynamically compress CoT at inference time by forcing the model to "disregard" steps predicted to be decorative, thereby improving latency? Or could it be used to enforce faithfulness in safety-critical applications?

---

> ### Author Response · Authors · 2025-11-21
>
> 1. > Practical Implications: While the analysis is fascinating (especially Section 7), the paper is light on the practical implications of its findings. The discovery of the "True Thinking direction" is a powerful result, yet the discussion of its applications is underdeveloped...Could the True Thinking Score (TTS) be adapted to serve as a more robust reward signal? ...
>
>
> Thank you for your comments. We acknowledge this work is relatively light on application as we focus on providing analysis and fundamental understanding of reasoning behaviors in LLMs. In Section 7.2, we explore the application of forcing the model to think through its correct reasoning traces of self-verification by steering along the TrueThinking direction. This enables the model to truly correct the errors in reasoning and improve its accuracy.
>
> We appreciate that your comments highlight several promising directions for future work.
>
> **Monitoring.** One could leverage the TrueThinking direction to build a latent monitor that compares intermediate hidden states to this direction and predicts, in real time,
> whether the model is truly engaging with a step before the final answer is produced.
> Future work could, for example, investigate which tokens to probe within the current reasoning step and which similarity metric (e.g., cosine similarity) is most effective for comparing hidden states to the TrueThinking direction.
>
> **Dynamic Compression at inference time.** We believe this is a very promising direction. A natural idea is to stop generation and elicit an early-exit answer once a step is detected as decorative. However, our results show that decorative-thinking and true-thinking steps are interleaved. This naive strategy may therefore be flawed, as it risks skipping later, important steps. Future work can build on our findings to develop more principled CoT compression methods that selectively truncate low-impact steps while preserving crucial reasoning.
>
> **Fine-tuning.** We plan to explore using the TrueThinking direction for training, inspired by [1].
> In particular, one could steer the model along the reverse TrueThinking direction while training it to correct or change its answer.
> Designing and analyzing such an adversarial training scheme is nontrivial and would likely require a dedicated follow-up paper.
>
> #### References:
>
> [1] Yu, L., Do, V., Hambardzumyan, K., & Cancedda, N. Robust LLM safeguarding via refusal feature adversarial training. Proceedings of the Thirteenth International Conference on Learning Representations, 2025.
>
>
> 2. > Limitation and Generalizability of the Method: The methodology appears to be designed for and evaluated on tasks with clearly demarcated, sequential, and often computational steps (i.e., mathematical reasoning)...
>
> Thank you for your very insightful questions. Apart from diverse mainstream math reasoning datasets, we also add experiments on CommonsenseQA,  a non-numeric and semantically diverse reasoning benchmark. Standard intervention schemes that rely on perturbing numerical values are not suitable in this setting. To apply interventions on such content, we prompt a separate large language model (GPT-5) to produce subtle, semantically coherent distortions of the initial reasoning step.  This makes perturbation applicable to different reasoning scenarios. We observe similar patterns to our results in math reasoning. Results are shown in Figure 10 in Appendix E. This supports the applicability of our framework to domains where reasoning is more open-ended, and harder to quantify. We view our methodology as a general tool that future work can readily apply to additional reasoning scenarios.
>
> However, we acknowledge that our framework is most straightforward when there is a verifiable answer, and applying it in settings with more subjective “truth” (e.g., open-ended dialogue) is more challenging. In such cases, one could instead use changes in perplexity over the entire output sequence to quantify the effect of perturbing a step, which still fits within our framework. In this work, we have evaluated our framework across multiple datasets and models and observed consistently strong behavior, suggesting that our approach is not tied to a single benchmark. We therefore expect our methodology to generalize beyond the tasks studied here, and future work can leverage our framework to explore different domains.

---

> > ### Author Response · Authors · 2025-11-21
> >
> > 3. > Interpretation of "Decorative" Steps: The finding that reasoning contribution is unevenly distributed is intuitive. However, the paper's binary classification of steps as "true" vs. "decorative" may be an oversimplification……. The paper would be strengthened by a more nuanced discussion of this spectrum, rather than a strong binary claim.
> >
> > We agree that it is inherently continuous as in our TTS framework, more "true" step means greater probability change after perturbation.  We did not explicitly threshold TTS to hard-label steps as “true” or “decorative”.  We use thresholding in Section 4 only to extract the TrueThinking direction. We have updated the writing to avoid misunderstanding.
> >
> > We use the terms “true thinking” and “decorative thinking” as shorthand to describe more faithful vs. less faithful reasoning behavior, mainly to make the phenomena easier to grasp for readers.
> > We analyze this spectrum in more detail in Appendix G.1, where we group steps by TTS ranges and use our steering tests as a testbed: steering directions extracted from higher-TTS ranges produce stronger and more consistent engagement effects, while lower-TTS ranges show weaker effects.
> >
> >
> > 4. > Computational Cost: The proposed TTS evaluation framework seems computationally prohibitive. It requires multiple perturbed forward passes for every step being evaluated within a CoT. This high cost could severely limit its practical utility for large-scale evaluation, real-time monitoring, or integration into a training pipeline.
> >
> > Thank you for your comments. We acknowledge that TTS is computationally expensive, since it requires multiple perturbed forward passes per step. However, our goal in this work is not to propose a real-time detector, but to introduce a theoretically grounded diagnostic for step-wise faithfulness. We need to, in the first place,  establish the finding that many steps (including “aha moments”) are in fact decorative and not causally used, even if the procedure is relatively costly.
> >
> >  To the best of our knowledge, we are the first to systematically measure faithfulness at the level of individual CoT steps across realistic benchmarks; prior work has primarily provided suggestive evidence of unfaithful CoT on carefully constructed or adversarial examples [1, 2], rather than quantifying where and how often unfaithful steps occur in typical reasoning traces.
> >
> > In practice, TTS can be applied on subsets of data rather than on every CoT trace, keeping the cost manageable for offline evaluation. Furthermore, one could leverage this TrueThinking direction to build a latent monitor that compares intermediate hidden states to this direction and attempts to predict, in real time, whether the model is truly engaging with a step before the answer is produced. Developing and validating such a predictive monitor is nontrivial and beyond the scope of this paper, but we view it as a promising avenue for future work.
> >
> > #### References:
> >
> > [1] Chen, Yanda, et al. "Reasoning Models Don't Always Say What They Think." arXiv preprint arXiv:2505.05410 (2025).
> >
> > [2] Arcuschin, Iván, et al. "Chain-of-thought reasoning in the wild is not always faithful." arXiv preprint arXiv:2503.08679 (2025).

---

> > > ### Comment · Reviewer_GQsZ · 2025-11-25
> > >
> > > Thank you for your rebuttal. I think this is an interesting work. Though there are still some potential practical limitations, this work provides a promising and fundamental analysis of evaluating the quality of reasoning steps. I will keep my score.

---

### Official Review · Reviewer_5Kcg · 2025-10-31

**Soundness:** 2
**Presentation:** 2
**Contribution:** 2
**Rating:** 4
**Confidence:** 3

**Summary:**

This paper aims to add another layer to the ongoing conversation about chain-of-thought reasoning by introducing the distinction between "true" and "decorative" thinking steps. They attempt to formalize this distinction using a causal framework, specifically the Average Treatment Effect (ATE) to determine which steps genuinely influence the final outcome and which are just superficial. There’s a certain incremental value in trying to categorize different types of reasoning steps. However, I have two major reservations.

First, the use of ATE in this context is questionable. While ATE is a standard tool in causal inference, applying it to the inner workings of a language model is far from straightforward. In a classical causal setting, we have well-defined interventions. Here, the notion of "intervening" on a reasoning step is much more nebulous, and this makes their causal analysis feel like a stretch: they’re trying to impose a rigorous framework on something that may not be well-defined enough to support it. It's possible to get into a discussion here, but I think that at a deeper level the discussion about appropriateness of the particular tool from the causal toolkit is a waste of time, because there's a second objection that might mean the whole causal setup is a waste of time anyway.

Second, and more fundamentally, the entire causal approach is a nonstarter because it labels reasoning steps based on the eventual correctness of the answer. This creates a backward-looking definition of "true" thinking that depends on the outcome rather than the inherent quality of the reasoning. As a result, the distinction between "true" and "decorative" thinking becomes arbitrary. For example, we might have a "Kahneman's Demon" waiting at the logits at the end of a model which may choose to change the final answer from correct to incorrect or vice-versa, based on an opaque functional dependence on the intermediate reasoning steps; in this thought experiment, there is by construction a functional/causal dependence on the final answer on the reasoning, but whether any reasoning is "true" or "decorative" is rendered arbitrarily, depending on the demon's actions at some later point in time. To drive the point home in a different way, if someone turned the computer off before the final answer were provided, the reasoning would be neither "true" nor "decorative" and instead have some indeterminate status, which is a strong smell of a bad definition (almost like the authors started reaching for mathematical busywork before thinking carefully about whether it was conceptually appropriate). These considerations go against any sensible definition of good reasoning which must take into account some kind of semantic condition on reasoning as a process, agreed upon by basically everyone forever (definitely not the hill that the authors want to die on). I think charitably this is just a case of "true"/"decorative" thinking being a bad name that suggests something stronger than what it actually is, and I'd be totally happy if that initial terminological choice were just fixed.

In sum, while there’s a glimmer of interest in trying to refine the taxonomy of reasoning steps, the methodological choices and conceptual foundations are too shaky to support the claims as they stand, but this might hinge solely on the choice of terminology (words mean things and unless the authors identify as recreational word users I think they would agree). I’d encourage the authors to reconsider their approach and either find a more suitable conceptual framework or tone down the claims. I will note however that this CoT paper was the most memorable out of the three CoT papers I've had to review this batch!

**Strengths:**

- Introduces an explicit causal framework (ATE-based) for analysing which CoT steps actually influence predictions.
- Attempts a mechanistic link between latent space dynamics and reasoning behaviour (“TrueThinking direction”).
- Empirically rich: multiple models, datasets, and steering tests.
- Conceptually provocative: raises the problem of “decorative reasoning,” a useful term for future critique.

**Weaknesses:**

- “True vs decorative” reasoning is outcome-defined and retrospective; conceptually half-baked
- Causal formalism is unconvincing: interventions and counterfactuals are ill-specified in MI generally
- The TrueThinking direction is under-motivated and likely an artefact of linear probing.
- Heavy reliance on visual examples and statistical noise; no formal significance testing.
- Overclaims about implications for “AI deception” and safety.

**Questions:**

- How robust are TTS and steering effects across random seeds or different tokenizations?
- Can “true thinking” be identified before observing the final answer?
- What exactly constitutes an intervention in hidden-state space—is it reproducible?

---

> ### Author Response · Authors · 2025-11-21
>
> 1. >  …, As a result, the distinction between "true" and "decorative" thinking becomes arbitrary. For example, we might have a "Kahneman's Demon" waiting at the logits at the end of a model which may choose to change the final answer from correct to incorrect or vice-versa... “True vs decorative” reasoning is outcome-defined and retrospective; conceptually half-baked
>
>
> Thank you for your comments. We would like to point out the definition is not based on whether the ultimate result is correct or not. Our definition quantifies the end-to-end causal contribution of the step to the ultimate result.
> The intuition is simple: if the verbalized reasoning step in CoT for computing the answer is faithful, then the model should truly think over it internally and thus the ultimate answer should depend on that step.
>
> We respectfully disagree with your thought experiment, which may not be applicable to LLMs.
> The hypothesis of "Kahneman's Demon" is interesting, while it hypothesizes that LLMs’ output can be arbitrary and irrelevant to the steps it relies on.
> Such separation between depending on a step and outputting the answer is unverified empirically.
>
> Practically, our evaluation examines how much the step is contributing to the model’s prediction. It indeed helps to identify useless decorative steps that consume tokens while do not contribute to the LLMs' predicted answer.
>
> 2. > In a classical causal setting, we have well-defined interventions. Here, the notion of "intervening" on a reasoning step is much more nebulous, and this makes their causal analysis feel like a stretch … Causal formalism is unconvincing: interventions and counterfactuals are ill-specified in MI generally’
>
> Thank you for your comments.
> We view replacing numbers in a reasoning step as a legitimate and well-defined intervention for LLMs.
> In classical causal inference, interventions are formal objects on abstract variables;
> their “well-definedness” comes from a precise operational specification of how the variable is set.
> In our case, the intervention is equally concrete:
> we define a binary treatment on a particular reasoning step by replacing the numerical values in a specific step with systematically perturbed values,
> while holding all other tokens fixed. Because the perturbed step has the same structure, position, and wording as the original (differing only in numeric content),
> it preserves the type of computation the model is encouraged to perform, making numeric replacement a natural intervention for probing whether a step is actually used in the model’s reasoning.
> We can indeed empirically find many cases where perturbed steps can barely change LLMs’ prediction.
> Such interventions have been widely employed to investigate CoT in past works [1, 2, 3].
> We see our use of causal formalism not as over-claiming a full structural model of the LLM, but as providing a principled language to formalize and extend existing CoT intervention practices.
>
> #### References:
>
> [1] Tamera Lanham, Anna Chen, Ansh Radhakrishnan, Benoit Steiner, Carson Denison, Danny Hernandez, Dustin Li, Esin Durmus, Evan Hubinger, Jackson Kernion, et al. Measuring faithfulness in chain-of-thought reasoning. arXiv preprint arXiv:2307.13702, 2023.
>
> [2] Yu, Xiangning, et al. "Causal Sufficiency and Necessity Improves Chain-of-Thought Reasoning." The Thirty-ninth Annual Conference on Neural Information Processing Systems, 2025.
>
> [3] Yee, Evelyn, Alice Li, Chenyu Tang, Yeon Ho Jung, Ramamohan Paturi, and Leon Bergen. Dissociation of Faithful and Unfaithful Reasoning in LLMs. In Proceedings of the Conference on Language Modeling (COLM 2024).

---

> > ### Author Response · Authors · 2025-11-21
> >
> > 3. > The TrueThinking direction is under-motivated and likely an artefact of linear probing.
> >
> > Thank you for your comments. We would like to point out that the TrueThinking direction is well-motivated and has been rigorously tested to ensure it is not an artefact.
> >
> > The TrueThinking direction is motivated to understand the reasoning behavior of LLMs in the internal space. Our findings suggest LLMs internally encode the variable of true thinking,
> > which is steerable to mediate the model’s reasoning behaviors.
> > Crucially, we validate this direction behaviorally. We design two causal tests to evaluate whether the TrueThinking direction can control the model’s internal engagement with a reasoning step and compare it with different baselines, i.e., random steering directions, directions extracted from past works’ metrics, and scaling attention.
> > Our identified direction demonstrates noticeable effects that are much higher than other baselines.
> > These results indicate that the TrueThinking direction captures a genuine, behaviorally meaningful aspect of the model’s internal reasoning, rather than a spurious artefact.
> >
> > 4. > Heavy reliance on visual examples and statistical noise
> >
> > Thank you for your review. The visual examples are intended only as illustrative cases to make the phenomena more interpretable;
> > all claims in the paper are supported by aggregated quantitative results.
> > We report the distribution of TTS and steering effects (flip rates) across multiple models and datasets, where the reported values are means over multiple random runs.
> > In our experiments, these effects are stable across runs, models, and datasets, indicating that they are not driven by isolated outliers.
> > As for “different tokenizations,” each model comes with its own tokenizer, and changing the tokenizer would amount to changing the model itself.
> > Our experiments on different models in our initial script show consistent results.
> >
> >
> > 5. > Overclaims about implications for “AI deception” and safety.
> >
> > Thanks for your comments. We would like to clarify that AI deception and safety are not the primary focus of this work, and we do not make any strong claims on these topics in the main text.
> > They are mentioned only briefly in the abstract, introduction, and conclusion as part of a broader discussion of potential implications.
> >
> > Prior work [1]  has proposed using CoT traces to monitor or “read off” a model’s internal decision process, which implicitly assumes that the CoT is faithful to the model’s inner reasoning.
> > Our results show that some steps are decorative (i.e., not used internally while being phrased as if they were), suggesting that naively trusting such traces for oversight could be risky.
> > We frame this as a possible concern and a direction for future work, not as a definitive claim. We have updated our writing to avoid the misunderstanding.
> >
> > [1] Baker, Bowen, et al. "Monitoring reasoning models for misbehavior and the risks of promoting obfuscation." arXiv preprint arXiv:2503.11926 (2025).
> >
> > 6. > Can “true thinking” be identified before observing the final answer?
> >
> > Thank you for highlighting this interesting direction. By design, our TTS framework is a post-hoc measure: it quantifies the causal effect of a step on the final answer, so it necessarily requires observing that answer. This is inherent to our definition of faithfulness in terms of causal contribution to the outcome. That said, our results suggest that “true thinking” is encoded in a latent direction in hidden states. In principle, one could leverage this TrueThinking direction to build a latent monitor that compares intermediate hidden states to this direction and attempts to predict, in real time, whether the model is truly engaging with a step before the answer is produced. Developing and validating such a predictive monitor is nontrivial and beyond the scope of this paper, but we view it as a promising avenue for future work.

---

> > > ### Author Response · Authors · 2025-11-21
> > >
> > > 7. > What exactly constitutes an intervention in hidden-state space—is it reproducible?
> > >
> > > Thank you for your question. The steering experiment is reproducible and has been widely implemented to interpret LLMs [1, 2, 3, 4].
> > > The intervention, as detailed in Section 4, adds a steering direction to the hidden states at some layer in LLMs.
> > > This will shift the representation of those hidden states and thus lead to different behaviors of LLMs.
> > > We will release the specific vectors to the public.
> > >
> > > #### References:
> > >
> > > [1] Samuel Marks and Max Tegmark. The geometry of truth: Emergent linear structure in large language model representations of true/false datasets. arXiv preprint arXiv:2310.06824, 2023.
> > >
> > > [2] Kenneth Li, Oam Patel, Fernanda Viégas, Hanspeter Pfister, and Martin Wattenberg. Inference-time intervention: Eliciting truthful answers from a language model. Advances in Neural Information Processing Systems, 36:41451–41530, 2023.
> > >
> > > [3] Andy Arditi, Oscar Obeso, Aaquib Syed, Daniel Paleka, Nina Panickssery, Wes Gurnee, and Neel Nanda. Refusal in language models is mediated by a single direction. Advances in Neural Information Processing Systems, 37:136037–136083, 2024.
> > >
> > > [4] Yu, L., Do, V., Hambardzumyan, K., & Cancedda, N. Robust LLM safeguarding via refusal feature adversarial training. Proceedings of the Thirteenth International Conference on Learning Representations, 2025.

---

> > > > ### Comment · Reviewer_5Kcg · 2025-11-26
> > > > **Nah.**
> > > >
> > > > Look, it's clear that you're measuring something, I just don't think that it deserves to be called a "True Thinking" score, because I think that words have nonrecreational usage. I'm not sure whether this is even worth spelling out in detail, but since these reviews are going in the public record it would be a shame not to savour the irony that the authors of a paper attempting to coin a "True Thinking" score have disagreed with a thought experiment on the basis that there isn't evidence that the hypothetical is true.
> > > >
> > > > To recap my contention in broad strokes, I think it's methodologically suspect to define "True Thinking" with respect to the final (i.e., ultimate) outcome, independently of the correctness of that outcome --- note, if that contention stands, it doesn't matter how much causal mathematics you want to throw at it, the foundational concept is just no good.
> > > >
> > > > The reference to Maxwell's demon clearly missed, so here is a simpler hypothetical comparison. Suppose I claim that I have a great method to measure the "TrueWinning" score of horses in a race, and you ask me what it is so that you can go win some bets. I share the secret with you: first I wait for the race to finish, and then I assign the winning horse a "TrueWinning" score of 1, and all the other horses a score of 0.
> > > >
> > > > Something is wrong, but what? I think the problem is that when you suggest that you can measure Winning or Thinking you seem to be referring to some property intrinsic to the horses or the tokens in the context of races or inference. However, since the measure is dependent on the actual realised outcome, the "trueness" or "decorativeness" of a step can potentially change without anything about the step, or the context immediately surrounding or preceding it, changing.
> > > >
> > > > To restate an example I wrote previously but in a more constructive way, one test of whether the measure TrueThinking is only dependent on knowledge that is accessible to us in the present is asking what would happen if the computer turned off midway through inference? The authors literally cannot answer this because they have conceded the point: the measure depends on the ultimate outcome in the future.
> > > >
> > > > I have read your responses to other reviewers, where you suggest that the approach is conceptually sound. I disagree. I would prefer to decrease my score in response to this unsatisfactory rebuttal, but I will instead keep it the same because nothing has materially changed about the paper: in your language, the authors TrueThinking score has no effect on the ultimate goodness of the paper.

---

### Official Review · Reviewer_3zkX · 2025-10-31

**Soundness:** 3
**Presentation:** 3
**Contribution:** 1
**Rating:** 4
**Confidence:** 4

**Summary:**

This paper investigates the faithfulness of Chain-of-Thought (CoT) reasoning in large language models (LLMs). Prior work has shown that LLMs can arrive at correct answers without fully verbalizing their reasoning, or that reasoning steps may be post-hoc justifications rather than reflecting true internal computation. The authors ask: to what extent do LLMs genuinely think through each step as verbalized in their CoT?

To address this, the paper proposes a method to measure step-wise causality, distinguishing between faithful "true-thinking" steps, which causally affect the model's output, and unfaithful "decorative-thinking" steps, which do not. The authors also introduce a TrueThinking vector in the latent space, which allows steering the model to reinforce or ignore specific reasoning steps. Finally, they show that even self-verification steps in CoT can be merely decorative and not genuinely used by the model to check its solution.

**Strengths:**

- The paper presents a simple yet effective extension of the Average Treatment Effect (ATE) by conditioning on context, enabling the capture of cases not addressed in previous work, such as the faithfulness of verification steps.
- Introduction of a vectorial direction representing "true thinking" in the model's latent space, which is a novel approach to probing internal reasoning.
- Experimental evidence shows that the TrueThinking vector improves the flip-rate, demonstrating that steering along this direction significantly affects model predictions compared to random directions, attention scaling, or step removal.
- The TrueThinking vector can be computed on one dataset and effectively applied to other datasets, indicating some generalization potential.

**Weaknesses:**

- The extensions of the Average Treatment Effect are limited to averaging existing scores conditioned on context; alternative ways to combine these scores are not explored, which may limit the expressiveness of the method.
- Steering is implemented solely by adding or subtracting the TrueThinking vector; the effect of using a multiplicative factor or other transformations is not investigated.
- Experiments are conducted on only three datasets focused on mathematical problems, which restricts the evaluation of the general applicability of the approach.
- The method is quite simple, suggesting potential for further exploration of alternative strategies to identify more effective steering directions.

**Questions:**

The main contributions are:

- Extending the Average Treatment Effect (ATE) with two complementary interventions conditioned on context $C$ (steps before step $s$): a necessity test $ATE(1) = P(y^*|C,s) - P(y^*|C,s')$ and a sufficiency test $ATE(0) = P(y^*|C',s) - P(y^*|C',s')$, with averaging yielding the True-Thinking Score (TTS).
- Computing a TrueThinking vector in the latent space between true-thinking and decorative-thinking steps, which can steer reasoning in new questions.
- Showing that true-thinking and decorative-thinking steps are interleaved in a CoT, and that self-verification steps can be decorative.

Questions / clarifications:

1. Regarding causal tests: is the steering vector optimal to weaken or reinforce a step? Could there be contamination from latent vectors of other steps or questions? How does averaging ensure only the decorative or true-thinking aspect is retained?
2. Is the steering vector simply added/subtracted, or is a coefficient applied to determine its strength?
3. How do you explain that removing a reasoning step produces much lower results than your method, even though it should yield a high flip rate if the step is causally related to the output?
4. In Figure 5, "Applying TrueThinking direction to a step increases the model's attention" appears weakly visible; how was the overall effect measured to support this claim?

Minor remarks (do not affect the score):

- Line 81: Some notations for Figure 2 are not introduced beforehand.
- Line 93: Why average? Is this the optimal way to combine the two scores?
- Line 328: Capitalize the first letter of the sentence.
- Line 335: Paragraph is unclear. What does $\Phi$ represent? Is $\beta$ the same as lines 253 and 292? What is its value here? This section could be further developed.

---

> ### Author Response · Authors · 2025-11-21
>
> 1. > The extensions of the Average Treatment Effect are limited to averaging existing scores conditioned on context;
> alternative ways to combine these scores are not explored, which may limit the expressiveness of the method.
>
>
> Thank you for your comments. We explored geometric mean in our early experiments, while produces similar distribution patterns to the arithmetic mean. We also attempted to use a weighted average. But it is unclear how to choose principled weights: there is no strong prior favoring (ATE(1)) over (ATE(0)) or vice versa, and tuning these weights on a validation set would introduce additional hyperparameters and substantial computational overhead without clear theoretical guidance.
>
> We therefore adopt the simple average to combine necessity and sufficiency: it treats both ATE(1) and ATE(0) equally and
> already demonstrates strong empirical behavior in our steering tests. We leave an exploration of such alternatives as a promising direction for future work.
>
>
> 2. > Experiments are conducted on only three datasets focused on mathematical problems, which restricts the evaluation of the general applicability of the approach.
>
> Apart from diverse mainstream math reasoning datasets, we have also added experiments on CommonsenseQA, a non-numeric and semantically diverse reasoning benchmark. We observe similar patterns to our results in math reasoning. Results are shown in Figure 10 in Appendix E. This supports the applicability of our framework to domains where reasoning is more open-ended. We view our methodology as a general tool that future work can readily apply to additional reasoning scenarios.
>
> Finally, we need to point out that we focus on math reasoning for two main reasons:(i) It is widely assumed that CoTs in math are especially faithful due to their procedural nature and difficulty [1],
> and (ii) These tasks frequently trigger “aha moments”, yet there has been limited understanding of whether such steps are genuinely used.
> Our results reveal that CoTs often contain unfaithful steps even in competition-level math reasoning, and show the common hypothesis may not be true.
>
> #### References:
> [1] Emmons, Scott, et al. "When chain of thought is necessary, language models struggle to evade monitors." arXiv preprint arXiv:2507.05246 (2025).
>
> 3. > Steering is implemented solely by adding or subtracting the TrueThinking vector; the effect of using a multiplicative factor or other transformations is not investigated. ..Is the steering vector simply added/subtracted, or is a coefficient applied to determine its strength?
>
>
> We do not use a multiplicative factor and set the coefficient as 1.  In early experiments, we tried using larger multiplicative factors and observed that they often push the model into a regime where outputs become gibberish. For example, the generation may repeat a single token. We can also possibly learn a coefficient matrix. However, this would introduce an additional case-specific hyperparameter, reduce comparability across settings, and add nontrivial computational overhead.
> We therefore adopt the simplest setting, which already yields clear and robust steering effects in our experiments.

---

> ### Author Response · Authors · 2025-11-21
>
> 4. > The method is quite simple, suggesting potential for further exploration of alternative strategies to identify more effective steering directions. Is the steering vector optimal to weaken or reinforce a step?
>
>
> Thank you for your comments. We agree that difference-in-means for extracting the steering direction is simple,
> but we view this as a strength rather than a limitation. The  difference-in-means construction is a standard and effective way to obtain steering directions in representation space, and it has been successfully used to extract diverse concepts in LLMs' latent space in prior work (e.g., [1,2,3,4]). Its success is closely tied to the approximately linear structure of LLMs' internal representations [5].
>
> Despite its simplicity, our approach yields clear and robust steering effects for true thinking:
> adding or subtracting the resulting direction weakens or reinforces the model’s engagement with a reasoning step.
> In fact, the effectiveness of such a basic linear operation itself provides evidence that “TrueThinking” is encoded in a largely linear subspace where
> addition/subtraction is sufficient to modulate it.
>
> As acknowledged in our limitations, we do not claim that this direction is globally optimal for weakening or reinforcing a step.
> Our primary contribution is to demonstrate the existence of a steerable TrueThinking direction for the first time,
> and to show that internal reasoning can be modulated via such directions.
> Designing more sophisticated procedures to search for provably optimal or higher-performing steering vectors (e.g., via optimization over our identified directions) is an exciting direction for future work,
> but orthogonal to the main goal of this paper.
>
> #### References:
>
> [1] Samuel Marks and Max Tegmark. The geometry of truth: Emergent linear structure in large language model representations of true/false datasets. arXiv preprint arXiv:2310.06824, 2023.
>
> [2] Kenneth Li, Oam Patel, Fernanda Viégas, Hanspeter Pfister, and Martin Wattenberg. Inference-time intervention: Eliciting truthful answers from a language model. Advances in Neural Information Processing Systems, 36:41451–41530, 2023.
>
> [3] Andy Arditi, Oscar Obeso, Aaquib Syed, Daniel Paleka, Nina Panickssery, Wes Gurnee, and Neel Nanda. Refusal in language models is mediated by a single direction. Advances in Neural Information Processing Systems, 37:136037–136083, 2024.
>
> [4] Yu, L., Do, V., Hambardzumyan, K., & Cancedda, N. Robust LLM safeguarding via refusal feature adversarial training. Proceedings of the Thirteenth International Conference on Learning Representations, 2025.
>
> [5] Jiang, Yibo, et al. "On the origins of linear representations in large language models." arXiv preprint arXiv:2403.03867 (2024).
>
> 5. > Could there be contamination from latent vectors of other steps or questions? How does averaging ensure only the decorative or true-thinking aspect is retained?
>
> Thanks for your comments. As discussed in our Response 4, prior work has shown that simple linear concept directions (including difference-in-means) are effective for steering and probing LLM representations.
> This is consistent with the view that LLMs encode many high-level concepts in approximately linear subspaces,
> so that hidden states tend to form clusters along directions associated with those concepts.
>
> In our case, TTS provides a step-level label (true-thinking vs. decorative) that lets us select the relevant clusters of hidden states.
> By taking the difference of the mean representations between high-TTS and low-TTS steps, the shared factors are canceled,
> and the remaining direction is the one most strongly aligned with the “true vs. decorative thinking” distinction.
> Importantly, we empirically validate this: steering vectors extracted using TTS mediate the model’s behavior substantially more effectively
> than vectors derived from prior metrics. This suggests that TTS is a suitable metric and that the dominant component of our direction is indeed tied to the true-thinking/decorative distinction rather than arbitrary noise from other steps or questions.
>
>
> 6. > How do you explain that removing a reasoning step produces much lower results than your method, even though it should yield a high flip rate if the step is causally related to the output?
>
> Thank you for your comments. We need to clarify that this baseline does not mean removing a reasoning step.
> As stated in Section 7 Paragraph "Comparison baselines'', DropStep means using the steering vector extracted based on metrics of past works.
> The much lower results than our identified TrueThinking direction suggest that their metrics are flawed and do not measure the faithfulness well.
>
>
> 7. > Minor remarks (do not affect the score): Line 81: Some notations for Figure 2 are not introduced beforehand. Line 328: Capitalize the first letter of the sentence.
>
> Thank you for your review. We have fixed them in our updated version.

---

> > ### Author Response · Authors · 2025-11-21
> >
> > 8. > In Figure 5, "Applying TrueThinking direction to a step increases the model's attention" appears weakly visible; how was the overall effect measured to support this claim?
> >
> > Thanks for your comments. We have changed the figure in the main text to make the effect more noticeable. We computed the average attention change after steering. When steering along the TrueThinking direction, the attention of the steered tokens will increase on average by 193% and steering along the reverse direction will decrease the attention by 32.3% across the layers.

---

### Official Review · Reviewer_y5rC · 2025-11-01

**Soundness:** 2
**Presentation:** 2
**Contribution:** 2
**Rating:** 4
**Confidence:** 3

**Summary:**

This paper introduces a True Thinking Score (TTS) to evaluate step-wise causality in chain-of-thought reasoning, revealing that LLMs interleave "true-thinking" steps (causally impactful) with "decorative-thinking" steps (minimal causal impact). The authors propose metrics based on Average Treatment Effect with context interventions and identify a "TrueThinking direction" in latent space that can steer whether models internally engage with reasoning steps.

**Strengths:**

- The paper presents the first work to measure faithfulness at the individual step level rather than treating CoT as monolithic. The distinction between conjunctive ("AND") and disjunctive ("OR") causal contributions is novel and conceptually sound.
- The experiments are well thought through. The use of context perturbations ( ATE(c=0) alongside ATE(c=1)) addresses a limitation in prior work and captures cases where steps provide alternative/verification pathways rather than strict necessity.

**Weaknesses:**

1. Validation circularity is unresolved.
- You use steering experiments to validate TTS, but you extract steering directions **from** TTS scores. Isn't this somewhat circular? There is no ground truth for what models "truly think." The steering experiments only show TTS correlates with steerability, but this doesn't prove TTS measures genuine internal reasoning. It could measure something else that happens to be steerable. If TTS is wrong about what constitutes true thinking, wouldn't your steering directions also be wrong?
- I appreciate that the paper acknowledges this in the appendix, but given that this is a major limitation, I think it should be discussed more in the main paper.

2. Task setup is limited.
- The experiments are only conduced on math reasoning on 7B-8B models. Unclear if findings generalize to other domains (code, commonsense reasoning) or larger models.
- While I understand mechanistic interpretability experiments can be time consuming and difficult to run in larger models, I think at least the task diversity can be improved.

3. Writing and presentation clarity can be improved.
- In the introduction, notations $s$, $C$, and $y$ are used without any definition.
- I find it hard to follow when"Conjunctive" and "Disjunctive" are introduced (L81-88). There needs more explanation and motivation of why you characterize CoTs into these two groups. Why did you decide to use "Conjunctive" and "Disjunctive"? Are there possibly other ways of characterizing CoTs?
- Too much experimental details without sufficient explanation in L89-96. For readers who don't have a background in causality, it is hard to follow when you start introducing necessity and sufficiency tests of ATE.
- I'd recommend making the introduction more concise and general, and save the experimental details for section 3.

**Questions:**

1. You insert "\nThe final result is" to probe predictions mid-CoT. But couldn't this prompt itself change how the model processes the reasoning? Have you validated that this probe doesn't create artifacts?
2. You acknowledge (Section 7.1) that there's no ground truth for whether a step is truly used internally. But then how can you validate that TTS actually measures faithfulness rather than something else (e.g., redundancy, information bottlenecks)? (related to weakness 1)

---

> ### Author Response · Authors · 2025-11-21
>
> 1. > There is no ground truth for what models "truly think." The steering experiments only show TTS correlates with steerability, but this doesn't prove TTS measures genuine internal reasoning.  It could measure something else that happens to be steerable.…You acknowledge (Section 7.1) that there's no ground truth for whether a step is truly used internally. But then how can you validate that TTS actually measures faithfulness rather than something else?
>
>
> Thank you for your comments. While it is inaccessible to interpret internal states directly, we can measure models’ causal behavior as indirect evidence: *what will happen if the model is truly thinking over the step versus when it does not*. Our steering experiments test this first principle objectively (a) Engagement Test: Can steering make the model think through a step in CoT it initially ignores? (b) Disengagement Test: can steering in the reverse direction make the model disregard a step internally that the model initially follows? Successful steering vectors on our causal tests mean the vectors can causally mediate whether the model engages with a step in its internal reasoning. This further suggests the metrics used to extract the vectors measure the internal reasoning in LLMs.
>
> Compared with other steering directions (e.g., random steering vectors, steering vectors extracted from past works’ metrics), our TrueThinking direction extracted based on TTS demonstrates significantly better performance in mediating LLMs' reasoning behaviors. This provides empirical evidence that TTS is closely tied to LLMs’ internal reasoning.
>
> Besides, TTS is conceptually sound and grounded in a causal framework. By construction, it is logically coherent: it measures the faithfulness of a step via its causal contribution, decomposed into conjunctive (“AND”) and disjunctive (“OR”) forms of causal influence. It directly formalizes the intuitive question of whether a step makes a difference to the model’s prediction. In contrast, prior faithfulness metrics are often justified heuristically and lack both a principled causal foundation and an explicit empirical verification of whether the corresponding directions truly control internal reasoning.
>
>
> 2. > Validation circularity is unresolved. You use steering experiments to validate TTS, but you extract steering directions from TTS scores. Isn't this somewhat circular?
>
> Thank you for your comments. We would like to clarify that our steering experiments are actually not circular.  The examples in the steering tests are **independent** of any faithfulness metrics or steering vectors. As discussed in response 1, the steering experiments are purely driven by the consequence of true-thinking behaviors, and the test examples are constructed based on that. This ensures our tests are fair for different steering vectors. Especially, we compare our TrueThinking direction with the direction extracted from the past work metric (DropStep) and show that our direction demonstrates stronger steering effects. This further rules out the circularity.
>
>
>
> 3. > If TTS is wrong about what constitutes true thinking, wouldn't your steering directions also be wrong?
>
> We agree with this. But empirically our experiments show noticeable performance of our identified TrueThinking direction. The fact that the experiment succeeded suggests TTs is correct.

---

> ### Author Response · Authors · 2025-11-21
>
> 4. > The experiments are only conduced on math reasoning on 7B-8B models. Unclear if findings generalize to other domains (code, commonsense reasoning) or larger models. While I understand mechanistic interpretability experiments can be time consuming and difficult to run in larger models, I think at least the task diversity can be improved.
>
> Thank you for your comments. Apart from 1.5 B and 7B models in our initial script, we have also added experiments on 14B model. As shown in Figure 11 in Appendix F, our findings hold from 1.5B to 14B models.
>
> Apart from diverse mainstream math reasoning datasets, we have also added experiments on CommonsenseQA,  a non-numeric and semantically diverse reasoning benchmark. We observe similar patterns to our results in math reasoning. Results are shown in Figure 10 in Appendix E. This supports the applicability of our framework to domains where reasoning is symbolic, open-ended, and harder to quantify. We view our methodology as a general tool that future work can readily apply to additional reasoning scenarios.
>
> Additionally, While prior work (e.g., [1, 2]) has demonstrated the existence of unfaithful CoT reasoning, these studies typically rely on **specifically crafted test cases**. In contrast, our work is the first to propose a step-wise evaluation framework that applies broadly to practical reasoning datasets. We hope this provides a useful foundation for future work, as our framework can be readily applied to a wide range of tasks and domains.
>
> #### References:
>
> [1] Chen, Yanda, et al. "Reasoning Models Don't Always Say What They Think." arXiv preprint arXiv:2505.05410 (2025).
>
> [2] Arcuschin, Iván, et al. "Chain-of-thought reasoning in the wild is not always faithful." arXiv preprint arXiv:2503.08679 (2025).
>
> 5. > Writing and presentation clarity can be improved.
>
> Thanks for your comments.  We have rewritten our introduction and Section 3 in our updated script.
>
> 6. >  Why did you decide to use "Conjunctive" and "Disjunctive"?
>
> We use conjunctive (“AND”) and disjunctive (“OR”) contributions because they are the two basic logical ways in which a step can affect the model’s prediction in context. Together, they capture complementary modes of influence: whether a step is required jointly with others, or sufficient on its own in combination with alternatives, which provide a more comprehensive evaluation of its causal role.  We have improved our writing in the updated version.
>
>
>
> 7. > You insert "\nThe final result is" to probe predictions mid-CoT. But couldn't this prompt itself change how the model processes the reasoning? Have you validated that this probe doesn't create artifacts?
>
> Thanks for your question. This probe is designed to prompt the model to reason over its current generated steps to predict an answer.  In our preliminary experiments, we repeated the probing with alternative suffixes such as "Therefore, the answer is" and "Final answer:". They produce similar accuracies. The chosen prompt also mirrors the phrasing that the model typically generates at the end of its own solution to summarize the answer. We speculate that models have been finetuned to summarize their answers in this fashion before release. In addition, past work has widely employed this method and has shown that it can reliably elicit the model’s intermediate answer (e.g., [1, 2, 3]).
>
> #### References:
> [1] Tamera Lanham, Anna Chen, Ansh Radhakrishnan, Benoit Steiner, Carson Denison, Danny Hernandez, Dustin Li, Esin Durmus, Evan Hubinger, Jackson Kernion, et al. Measuring faithfulness in chain-of-thought reasoning. arXiv preprint arXiv:2307.13702, 2023.
>
> [2] Sree Harsha Tanneru, Dan Ley, Chirag Agarwal, and Himabindu Lakkaraju. On the hardness of faithful chain-of-thought reasoning in large language models. arXiv preprint arXiv:2406.10625, 2024.
>
> [3] Yichao Fu, Xuewei Wang, Yuandong Tian, and Jiawei Zhao. Deep think with confidence. arXiv preprint arXiv:2508.15260, 2025.

---

### Author Response · Authors · 2025-12-03
**General Response**

Dear Reviewers, AC, and SAC,

We are encouraged that reviewers recognized that our work (i) `addresses a critical and timely question` (GQsZ) and covers `cases not addressed in previous work` (3zkX, y5rC); (ii) is `novel and sound` (GQsZ, 3zkX, y5rC) and `empirically rich` (5Kcg); (iii) introduces a method that is `simple but effective` (3zkX); and (iv) provides a `mechanistic explanation of reasoning behaviors` (3zkX, 5Kcg, GQsZ) that has `generalization potential` (3zkX) and offers `compelling evidence for the paper’s claims and opens up new avenues for model interpretability and control` (GQsZ).

Regarding the concern on generalizability, we have shown that our approach extends to less formal and more open-ended non-numerical reasoning, where we observe consistent patterns of unfaithful reasoning steps (Appendix E in our updated script). We view our methodology as a general diagnostic tool that future work can readily apply to additional reasoning domains and tasks.

Moreover, we for the first time try to interpret unfaithful CoT internally. We reveal that LLMs may encode the very concept of faithful reasoning in latent space. We can use a steering direction extracted from our TTS to mediate whether LLMs use or disregard specific reasoning steps in CoT.

---

### Meta-Review · Area_Chair_1ni1 · 2026-01-10

**Summary:**

This paper proposes a step-level causal framework for assessing faithfulness of Chain-of-Thought reasoning in LLMs by quantifying each step's causal contribution to the model's final answer via context-conditioned interventions, identifying a True Thinking Score (TTS). It further extracts a latent "TrueThinking direction" used to steer whether models internally engage with particular reasoning steps, and argues that even self-verification "aha moments" can be decorative.

The reviewers generally agree that the work tackles an important interpretability/safety-motivated question and attempts to move beyond full trace analyses towards step-level diagnostics. The steering results are interesting from a mechanistic point of view and the paper reports cross-dataset transfer.

Reviewer concerns are centered around the conceptual framing of the "true vs decorative" notion; validation with TTS; generality. More details below.

**Reviewer Concerns:**

The additional experiment with CommonsenseQA helps with concerns around generality, but it is limited.

Conceptual and causal framing concerns remain unresolved for multiple reviewers. The disagreement about whether the proposed "true vs decorative" notion being well-defined would probably not be settled if there was additional reviewer discussion.

There is also a validation concern regarding the reliance of TTS on steering results derived from labels produced by the same metric; this seems only partially addressed but it doesn't fully close the loop for a skeptical reader, for example through a decisive ablation.

Finally, while the rebuttal adds some discussion and at least one non-math benchmark, the broader applicability and practical limitations of the method remains somewhat limited.

**Reviewer Scores:**

Reviewer 5Kcg explicitly states they remain unconvinced post-rebuttal and will keep their score even if they considered lowering it.

There are unresolved issues highlighted by some reviewers that were missed by others (more so than paper strengths that subsets of reviewers identified and others missed). This, if there was more discussion, could counter-act any limited increase in scores by individual reviewers, and I would expect the overall distribution of scores to stay similar, other than reviewer y5rC potentially increasing their score by 1 thanks to the additional experiment.

---

### Decision · Program_Chairs · 2026-01-26

Reject